

# Observations of Tropical Tropopause Layer clouds from a balloon-borne lidar

Thomas Lesigne[1], François Ravetta[1], Aurélien Podglajen[2], Vincent Mariage[1], and Jacques Pelon[1]

[1]Laboratoire Atmosphères, Observations Spatiales (LATMOS/IPSL), Sorbonne Université, UVSQ, CNRS, Paris, France
[2]Laboratoire de Météorologie Dynamique (LMD/IPSL), École Polytechnique, Institut Polytechnique de Paris, Sorbonne Université, École Normale Supérieure, PSL Research University, CNRS, Paris, France

**Correspondence:** Thomas Lesigne (thomas.lesigne@latmos.ipsl.fr)

**Abstract.**

Tropical Tropopause Layer (TTL) clouds have a significant impact on the Earth's radiative budget and regulate the amount of water vapor entering the stratosphere. During the Strateole-2 observation campaign, three microlidars were flown onboard stratospheric superpressure balloons from October 2021 to late January 2022, slowly drifting only a few kilometers above the TTL. These measurements have unprecedented sensitivity to thin cirrus and provide a fine-scale description of cloudy structures both in time and space. Case studies of collocated observations with the space-borne lidar Cloud-Aerosol Lidar with Orthogonal Polarization (CALIOP) show a very good agreement between the instruments and highlight the unique ability of the microlidar to detect optically very thin clouds below CALIOP detection capacity (optical depth $\tau < 2 \cdot 10^{-3}$). Statistics on cloud occurrence show that TTL cirrus appear in more than 50 % of the microlidar profiles and have a mean geometrical depth of 1 km. Ultrathin TTL cirrus ($\tau < 2 \cdot 10^{-3}$) have a significant coverage (16 % of the profiles) and their mean geometrical depth is below 500 m.

## 1 Introduction

In the Tropics, the transition between troposphere and stratosphere occurs in a vertically extended layer (14 to 18.5 km) sharing characteristics from both troposphere and stratosphere: the Tropical Tropopause Layer (TTL, Fueglistaler et al. (2009); Randel and Jensen (2013)). Most of the air entering the stratosphere makes its way through the TTL along the ascending branch of the Brewer-Dobson circulation (Brewer, 1949). TTL is then often referred to as the "gate to the stratosphere". On their way up, air masses encounter extremely low temperatures at the cold point tropopause (CPT, $\sim 17$ km, $\sim 190$ K) that freeze-dry a great part of their water content and are ultimately responsible for the dryness of the lower stratosphere (Holton et al., 1995). Although water vapor concentration in the stratosphere is very low ($\sim 5$ ppmv), it has a significant radiative impact on the whole climate system (Solomon et al., 2010) and plays a major role in stratospheric chemistry (Fueglistaler et al., 2009). Yet its evolution is not accurately represented in today's climate models. It is thus necessary to get a better understanding of the various TTL processes (transport, dynamical, radiative and microphysical processes) modulating the amount of water vapor and other trace gases that eventually reaches the stratosphere.



At the heart of the interplay between those processes, TTL clouds have been subject to numerous studies in the past decades. Thanks to their high vertical resolution and unique sensitivity to tenuous clouds, lidar observations have long been used to characterize tropical clouds, operated from the ground (e.g., Platt et al., 1984, 1987), or from research vessels (e.g., Fujiwara et al., 2009), but their spatial coverage is limited and they suffer from being potentially blinded by opaque clouds between the ground and the upper troposphere. Passive space-borne observations (either radiometers (e.g., Prabhakara et al., 1988) or solar occultation measurements (e.g., Wang et al., 1994)) have broaden the picture, providing almost global observations, but they lack sensitivity and resolution to fully capture the TTL cloud coverage. Since the pioneer mission Lidar In-space Technology Experiment (LITE, Winker and Trepte, 1998), space-borne lidars have overcome these limitations. For the past 17 years, the Cloud and Aerosol Lidar with Orthogonal Polarisation (CALIOP) onboard the Cloud-Aerosol Lidar and Infrared Pathfinder Satellite Observation (CALIPSO) provided continuous observations that fed a great deal of cloud studies (Yang et al., 2010; Martins et al., 2011; Iwasaki et al., 2015). This mission recently ended on August 1, 2023. CALIOP's cloud observations have intensively been evaluated against other types of measurements, from ground-base lidars (Thorsen et al., 2011) to geostationary weather satellite (Sèze et al., 2015). Recent airborne campaigns such as the NASA Airborne Tropical TRopopause EXperiment (ATTREX, Jensen et al., 2017) have enabled the in situ characterization of thin TTL cirrus. A noteworthy result from aircraft data was the characterization of a systematic relationship between TTL clouds and equatorial and gravity waves (Kim et al., 2016). This finding was later confirmed with space-borne (Chang and L'Ecuyer, 2020) and more recently with balloon-borne observations (Bramberger et al., 2022).

Long-duration stratospheric balloon constitute an invaluable platform to better characterize clouds distribution. Since the balloon is slowly drifting with the air, it is able to capture the fine scale spatial variability of the underlying cloud scene. Here, we introduce the first observations from the Balloon-borne Cirrus and convective overshOOt Lidar (BeCOOL, Ravetta et al., 2020, 2023). BeCOOL nadir-looking lidar has a viewing geometry comparable to CALIOP, but benefits from a significantly higher signal to noise ratio (SNR) in the TTL and upper troposphere. BeCOOL was recently flown for the first time onboard three superpressure balloons (SPBs) in the framework of the Strateole-2 project (Haase et al., 2018; Corcos et al., 2021; Bramberger et al., 2022). The SPBs were launched from Seychelles Island and travelled up to the middle of the pacific ocean at about 20.5 km (50 hPa) between October 2021 and January 2022, gathering 700 nighttime hours of high-resolution lidar profiles.

The article is organised as follow: Section 2 presents the different data sets and the cloud classification. In Sect. 3, three case studies of collocated BeCOOL/CALIOP observations are analyzed to contrast the two instruments, their sampling and detection capability. Section 4 is dedicated to a statistical description of the balloon-borne cloud data and a statistical comparison with space-borne lidar data. Conclusions and perspectives are in section 5.





## 2 Lidar Data sets

### 2.1 Balloon-borne lidar data

BeCOOL nighttime observations have been gathered during the first Strateole-2 fiels campaign, from 20 October 2021 to 26 January 2022. Three balloons carrying the lidar were successively launched from Seychelles Islands and drifted eastward at about 20 km of altitude near the equator, their trajectories are shown on Fig 1.a. The main characteristics of the flights are presented Table 1 and a summary of the technical specifications of BeCOOL microlidar are presented in Table 2. A comprehensive description of the microlidar and the processing of Level 1 and Level 2 products (respectively attenuated backscatter profiles and the retrieved geometrical and optical properties of cloud and aerosol layers) can be found in Ravetta et al. (2023). The Level 2 data set has been retrieved after averaging 10 consecutive 1-minute profiles in order to improve the SNR thus the detection of optically thin clouds.

**Table 1.** Main characteristics of the three Strateole-2 flights carrying BeCOOL microlidar; $\overline{z}$ is the mean altitude above sea level and $\overline{|\mathbf{u}|}$ the mean ground speed of the balloon.

| Flight | Strateole-2 ID | Launch Date | End Date | $\overline{z}$ | $\overline{|\mathbf{u}|}$ | Number of 1 minute profiles |
|---|---|---|---|---|---|---|
| 1 | ST2_C1_02_STR1 | 2021-10-20 | 2021-11-01 | 20.2 km | $11.2\,\mathrm{m\cdot s^{-1}}$ | 3573 |
| 2 | ST2_C1_08_STR1 | 2021-11-05 | 2021-12-29 | 20.3 km | $7.2\,\mathrm{m\cdot s^{-1}}$ | 15617 |
| 3 | ST2_C1_13_STR1 | 2021-11-15 | 2022-01-25 | 20.3 km | $6.4\,\mathrm{m\cdot s^{-1}}$ | 19742 |

Figure 1 shows the trajectories of the three flights and the lidar curtains (time *vs* altitude) of attenuated backscatter and reveals a large variety of different cloud scenes. Intense surface echos (i.e., the ocean or exceptionally the land surface) are seen in 86 % of the profiles. The lidar beam is fully attenuated by opaque clouds otherwise. In profiles reaching the surface, the ubiquitous aerosol-rich boundary layer generally occupies the lowest 2.5 km along with frequently occurring low-level clouds (cumulus and stratocumulus). Geometrically thin (a few hundred meters), horizontally extensive mid-level clouds are often found above, below 10 km, mainly between 5 and 8 km; they typically have large backscatter and are likely pure liquid or mixed-phase clouds. Above 10 km are pure ice clouds, cirrus and deep convective clouds. The clouds' vertical structure can be fully resolved up to an optical depth $\tau_{max} \simeq 3$, a threshold value depending on the energetic conditions and optical efficiency of the instrument which both vary with thermal conditions onboard the gondola. Clouds thicker than this appear opaque (the lidar beam is fully attenuated before reaching the bottom of the cloud) and only their upper part can be resolved. Typically, deep convective clouds have a large vertical extent that cannot be accurately inferred from BeCOOL observations.

For the purpose of this study, the area covered by the balloons has been zonally divided into three regions: Indian Ocean (55° E to 95° E), Maritime Continent (95° E to 165° E) and Central Pacific Ocean (165° E to 230° E). There is a striking contrast between very cloudy profiles over Indian Ocean and Maritime Continent, with frequent deep convection, and clear-sky conditions over the Central Pacific Ocean, in the second part of flights 2 and 3.



We built a classification of cloud profiles for the BeCOOL dataset. Clouds are first classified using a set of threshold values on their top and base altitude. Those altitudes have first been determined to retrieve cloud's optical properties where the lidar signal departs from the molecular background signal. For the cloud classification, in order to focus on the main part of clouds, top and base altitudes are slightly modified so that 15 % of the cloud optical depth lay above the new top altitude, and 15 % below the new base altitude. Cirrus clouds are here defined as all clouds with a base altitude lying above 10 km (i.e., temperatures below the glaciation threshold of supercooled droplets at about 240 K), then sub-classed as TTL cirrus if their base altitude is over 14 km. Convective clouds are here defined as opaque clouds (totally attenuating the lidar beam) with top altitude laying above 10 km. Mid-level clouds have a top altitude between 5 and 10 km. A last class gathers clouds that do not fit previous requirements, with top altitude above 10 km and base altitude below, sharing characteristics from both cirrus and mid-level clouds. This classification is somewhat restrictive in the case of deep convection, since mostly the core of convective cells will be flagged in this category, while a large part of the convective anvils will be classified as cirrus as long as BeCOOL's lidar beam goes through and the cloud base is above 10 km.

A profile classification has been built from this cloud classification. "Clear sky" is defined as profiles with no detected cloud above 5 km, since low level clouds and the planetary boundary layer are not considered in this study. "Deep convection" gathers profiles exhibiting any convective clouds, regardless of the presence of cirrus on top of it. "Cirrus only" and "Mid-level clouds only" stand for profiles where only such type of clouds are detected above 5 km. A last class, "Mixed multilayered scenes" gathers the other profiles, usually exhibiting complex overlay of cirrus and mid-level clouds.

Further classification of cirrus layers is performed based on their optical depth: thin cirrus have an optical depth below 0.1, which is about the detection lower bound for passive radiometer (McFarquhar et al., 2000), sub-visible clouds have an optical depth $\tau < 3 \cdot 10^{-2}$, a classical value from Sassen et al. (1989), and ultrathin cirrus have an optical depth $\tau < 2 \cdot 10^{-3}$ (which is about the detection threshold of CALIOP, see Sect. 4).

## 2.2 Space-borne lidar data

BeCOOL is compared to CALIOP space-borne lidar using the Level 2 Cloud and Aerosols merged product with a 5 km horizontal resolution, version 4.21 (Young et al., 2018). This data set reports optical and geometrical properties of detected clouds or aerosol layers along the satellite track. When CALIOP lidar curtains are displayed, the figures are generated using the Level 1 version 4.11 attenuated backscatter product (Kar et al., 2018). In this study, only the 532 nm channel is used. The main technical specifications of CALIOP lidar, along with BeCOOL's, are presented in Table 2.

While CALIOP flies at $\sim 8$ km·s$^{-1}$, achieving its native 1/3 km horizontal resolution from a single lidar shot, BeCOOL flies one thousand time slower, which allow to integrate individual lidar shots for a whole minute, considerably enhancing the SNR. This speed difference also implies that CALIOP provides an almost instantaneous description of cloudy structures at synoptic scale, while temporal and spatial evolution of the underlying scene are entangled in BeCOOL's observations. BeCOOL's laser divergence (667 μrad) is significantly higher than CALIOP's (100 μrad), meaning that BeCOOL's high SNR in near field decreases toward the surface due to geometric power dilution, whereas this effect can be neglected for CALIOP.





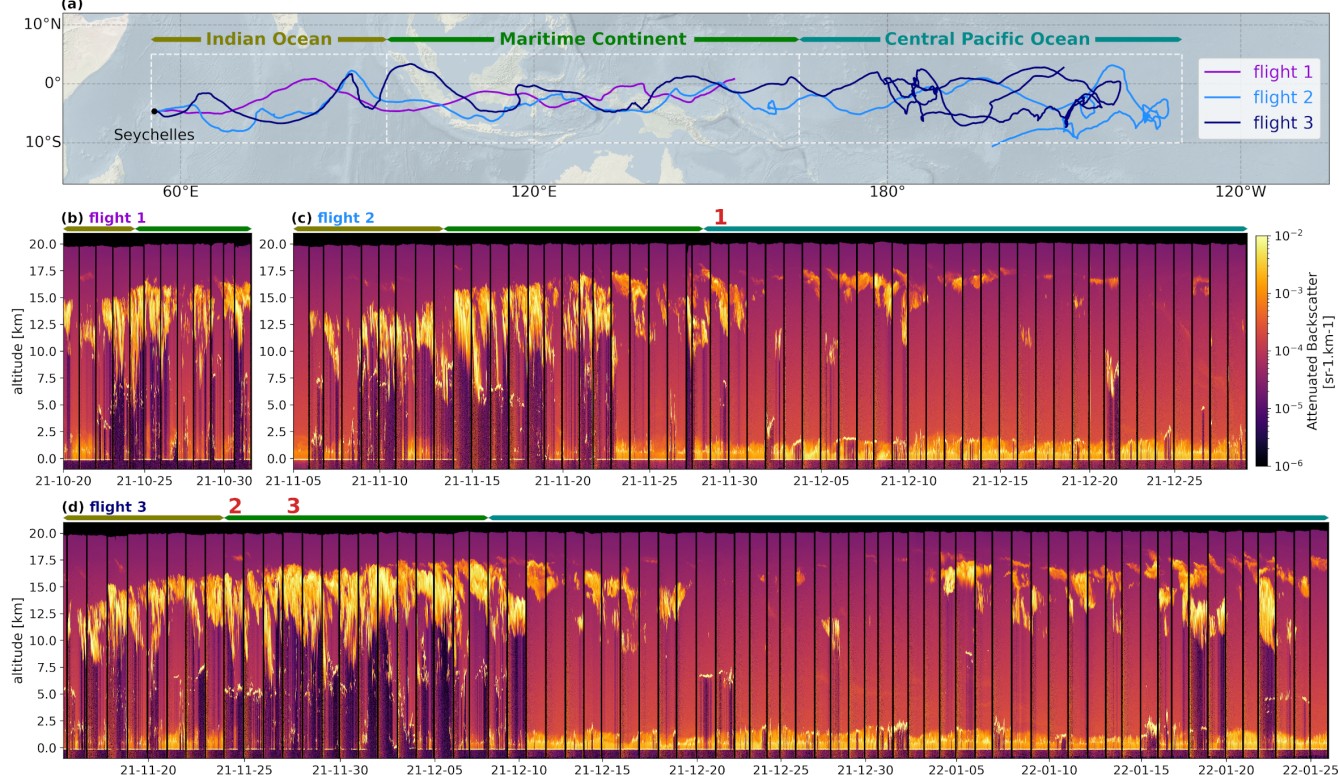

**Figure 1. a**: trajectories of the three balloons carrying BeCOOL during the first Strateole-2 scientific campaign, dashed white boxes show the three studied regions; **b-d**: lidar curtains (time *vs* altitude, attenuated backscatter) for the three flights, concatenated nights of observation (apart from thin vertical black lines, daytime has been removed for the sake of readability). Overflown regions are color-coded on top of the curtains. Red numbers (1, 2 and 3) highlight the three case studies presented in section 3.

Following Reagan et al. (2002), we assume that the optical depths retrieved at 808 nm (for BeCOOL) and 532 nm (for CALIOP) wavelengths are comparable, i.e. that the scattering particles are larger than $5 - 8$ μm such that there is only weak wavelength dependency of Mie scattering.

115      During the campaign, CALIOP was crossing the equator around 2:30 local time. Originally crossing the equator at 1:30 local time as part of the Afternoon-Constellation (A-train), CALIPSO was moved to a lower orbit in 2018 to join Cloudsat (Braun et al., 2019). As CALIPSO's fuel reserves were coming to an end, the satellite has been experiencing an orbital drift which explains the 2:30 AM crossing time during the campaign instead of the usual 1:30 AM.

## 3    Case studies of BeCOOL/CALIOP collocated observations

120      Three case studies of collocated BeCOOL/CALIOP measurements are now presented in order to compare the two instruments at coincidence time and highlight their complementarity due to the fundamental differences mentioned in the previous section.



**Table 2.** Overview of the main characteristics of BeCOOL and CALIOP lidars.

|  | BeCOOL | CALIOP |
|---|---|---|
| wavelenght | 808 nm | 532 nm* |
| pulse repetition rate | 4700 Hz | 20 Hz |
| pulse energy | 10 µJ | 110 mJ |
| depolarization channel | no | yes |
| altitude | 20 km | 700 km |
| ground speed | 0 to 25 $\mathrm{m \cdot s^{-1}}$ (mean of 7 $\mathrm{m \cdot s^{-1}}$) | 8 $\mathrm{km \cdot s^{-1}}$ |
| horizontal resolution Level 1 | 0 to 1.5 km (mean of 420 m) | 333 m |
| number of lidar shots | $\sim 3 \cdot 10^5$ | 1 |
| temporal resolution | 1 min | 5 ms |
| horizontal resolution Level 2 | 0 to 15 km (mean of 4.2 km) | 5 km |
| number of lidar shots | $\sim 3 \cdot 10^6$ | 15 |
| temporal resolution | 10 min | 0.7 s |
| vertical sampling | 15 m | 30 m below 8.2 km a.s.l. |
|  |  | 60 m above 8.2 km a.s.l. |
| laser beam divergence | 667 µrad | 100 µrad |
| diameter of the illuminated spot |  |  |
| 17 km a.s.l. | 2 m | 70 m |
| surface | 14 m | 70 m |

*CALIOP's 1064 nm channel has a low SNR and is not used in this study

The case studies correspond to different cloud scenes: a thick (anvil) cirrus, a thin cirrus and deep convection. To contextualize the cloud scene around lidar observations, we use the NOAA/NCEP GPM_MERGIR brightness temperature data in the atmospheric window ($\sim$11 µm). This product combines observations from 4 geostationary satellites and provides a global coverage with spatial resolution of 4 km and temporal resolution of 30 min (Janowiak et al., 2001).

Figures 2, 3 and 4 present lidar curtains from the two instruments along with the GPM-MERGIR brightness temperature map closest to the coincidence time, with a size of 5°×5°. In order to display lidar curtains from both instruments at a comparable resolution, CALIOP's horizontal resolution as been downgraded to 1 km below 8.2 km a.s.l., which is the native resolution above. The step from 30 to 60 m vertical resolution at 8.2 km a.s.l. is still visible on CALIOP's curtains. BeCOOL's curtains are displayed for the whole nights (roughly 11 h of observations), starting around 18:00 local time and ending around 5:30 local time, covering an horizontal distance of 200 to 500 km along the balloon's track depending on the wind speed. Each CALIOP's curtain is 570 km long (dashed line on the maps), covered in about 82 s.

As previously stated, CALIOP's observations are almost instantaneous and can be compared with a single brightness temperature map from GMP-MERGIR, latitude appearing then as a natural coordinate for CALIOP, while time stays the natural





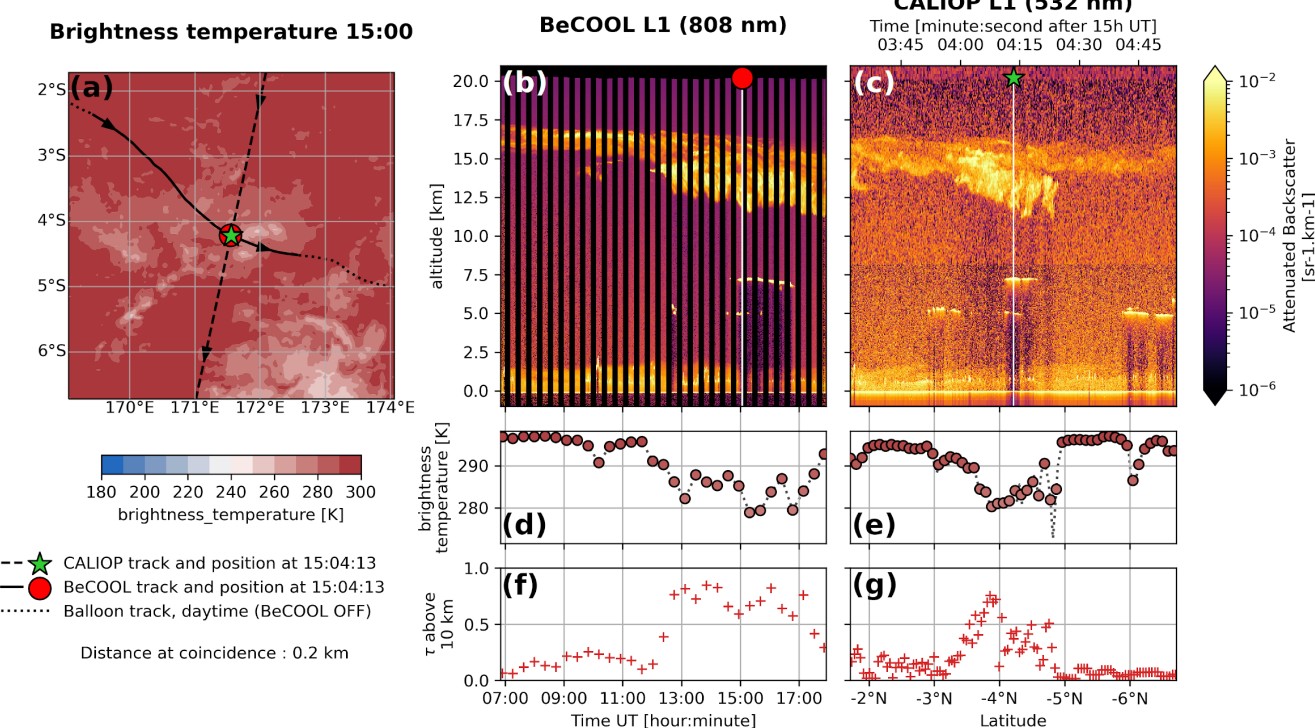

**Figure 2.** First case study: thick cirrus cloud, 2021-11-29. **a**: 11 μm brightness temperature map at 15:00 UTC; **b**: BeCOOL L1 curtain (along the solid line on the map); **c**: CALIOP L1 curtain (along the dashed line on the map); **d, e**: time series of brightness temperature under the balloon and the satellite; **f, g**: time series of optical depth $\tau$ above 10 km retrieved from BeCOOL and CALIOP.

coordinate for BeCOOL, which can only be compared with successive brightness temperature maps. Hourly maps for the three case studies are presented in the appendix, Fig. A1, A2 and A3.

### 3.1   First case study: thick cirrus cloud

Figure 2 shows an excellent coincidence that happened on 29 November 2021 over the Pacific Ocean (∼4° S, 172° E) for the second BeCOOL flight. The satellite track crossed the balloon track less than 1 km away from it. The balloon covered 430 km during this night. There is a perfect agreement between the two lidars at the coincidence time: they both capture a thick cirrus cloud extending from 12 to 16 km over two mid-level clouds with very small vertical extent around 5 and 7 km. CALIOP's curtain shows that at coincidence time this thick cirrus is embedded in a larger scale thinner laminar cirrus extending vertically from 14 to 16 km and horizontally all along the 570 track displayed here. The brightness temperature map at 15:00 UTC reveals the horizontal structure of this thick cirrus, centered on the coincidence spot and with an apparent radius of ∼100 km. BeCOOL's curtain and the hourly brightness temperatures maps on Fig. A1 allow to follow the temporal evolution of the scene under the balloon: from the beginning of the night up to 13:00 UTC, a thin and laminar cirrus vertically extending between 15



and 17 km, with an optical depth of 0.2 is overflown, then this cirrus thicken to extend vertically from 12 to 16 km, reaching an optical depth of 0.7-0.8. The balloon follows the thick cloud for the second part of the night as they are both advected eastward.

Brightness temperature (BT) below both instruments (Fig. 2.d-e) exhibits high values (almost 300 K) above the thin part

of the cloud, between 7:00 and 9:00 UT on Fig. 2.b and between 2 and 3° S on Fig. 2.c. BT drops down to 280 K above the thicker part of the cloud, after 13:00 UT on Fig. 2.b and around 4° S on Fig. 2.c. Cloud's contribution to upward thermal flux increases with optical depth, actually lowering the flux and revealing the thermal contrast between low temperatures at cloud level and higher temperatures below. BT (Fig. 2.d-e) and total cloud optical depth above 10 km (Fig. 2.f-g) are thus quite anti-correlated: $r = -0.88$ along BeCOOL's track, $r = -0.72$ along CALIOP's. These correlations would be more significant

without the presence of mid-level clouds around 5 and 7 km, that are not accounted for in the total cloud optical depth above above 10 km but further lower the BT (e.g., Fig. 2.e at 6° S).

At coincidence, the retrieved cirrus' optical depths are 0.6 for BeCOOL and 0.4 for CALIOP. Beyond the optical depth's spectral dependency, which is expected to be a second order effect, there is about 20 % of uncertainties in both retrieval, related to the assumed extinction-to-backscatter ratio (lidar ratio) and multiple scattering effects (Ravetta et al., 2023). In this thick

cirrus case, multiple scattering is the main source of uncertainty. Thus the two retrieved optical depth values can be considered in a fair agreement.

### 3.2 Second case study: thin cirrus clouds

The second case study (Fig. 3) happened for the third BeCOOL flight off the east coast of Sumatra island, Indonesia ($\sim 3°$ N, 97° E; distance at coincidence: 10.4 km) and corresponds to collocated observations of a very thin TTL cirrus, which is only

partially reported in CALIOP Level 2 data. BeCOOL curtain on Fig. 3 reveals clearly a thin cloudy layer above 17.5 km, first fading out from the beginning of the night until 14:00 then appearing back from 15:30 UTC and slowly thickening until 20:00 UTC reaching up to about $10^{-2}$ optical depth. This horizontally homogeneous, geometrically and optically very thin cirrus layer appears to fit in the description of Ultrathin Tropical Tropopause Clouds (UTTCs) reported by Peter et al. (2003). In CALIOP's curtain, this cloud can be identified by the human eye around 17.5 km in the 532 nm total attenuated backscatter

(Fig. 3.c). However, it is only reported in CALIOP L2 for about 10 s around coincidence. It was detected after an horizontal averaging of 80 km, the last step of the algorithm designed to improve SNR in order to detect tenuous features, but its horizontal extent could likely have been better constrained with even more extensive horizontal averaging. Given the limitation of CALIOP L2 algorithm for such case and for the sake of a fair comparison of the instrument capabilities, we manually retrieved the UTCC optical depth from CALIOP L1. We first improve the SNR applying an horizontal rolling mean over a 80 km window. Then,

we retrieve the cirrus optical depth at three latitudes: 4.5°, 3° and 1.5° N, keeping the same lidar ratio (21.8 sr) and multiple scattering factor $\eta$ (0.77) as reported in CALIOP L2 for the central part of this cloud. This cirrus' optical depth is $8.3 \cdot 10^{-4}$ at 5° N, increasing to $4.0 \cdot 10^{-3}$ at 3° N (coincidence) then decreasing to $2.2 \cdot 10^{-3}$ at 1.5° N. At coincidence, using the same assumed lidar ratio value for BeCOOL but keeping its own multiple scattering factor $\eta$ of 0.5 gives an new optical depth of $3.8 \cdot 10^{-3}$. Assuming the same lidar ratio, the optical depths at coincidence are thus in excellent agreement. The difference

between the optical depth of $10^{-2}$ reported in BeCOOL Level 2 at coincidence and the new optical depth value of $3.8 \cdot 10^{-3}$



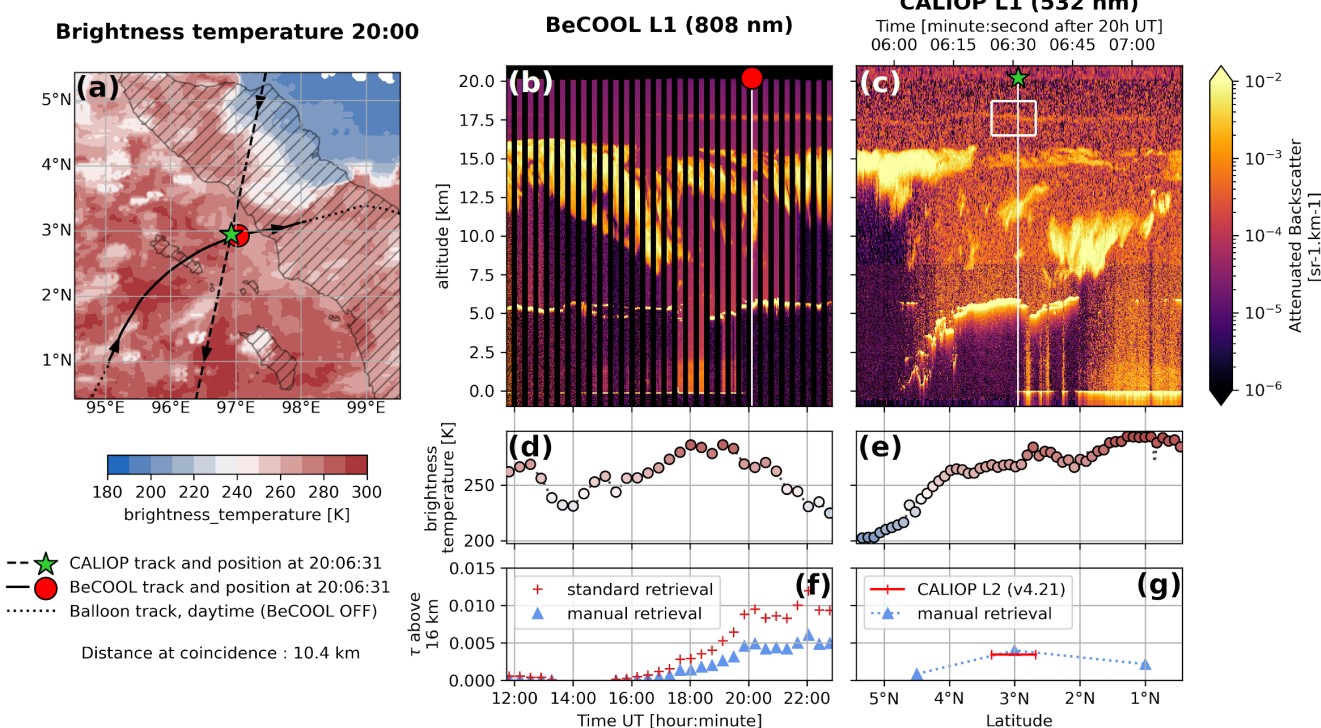

**Figure 3.** Second case study: thin cirrus cloud, 2021-11-24. **a**: 11 μm brightness temperature map at 20:00 UTC; **b**: BeCOOL L1 curtain (along the solid line on the map); **c**: CALIOP L1 curtain (along the dashed line on the map); **d**, **e**: time series of brightness temperature under the balloon and the satellite; **f**, **g**: time series of optical depth $\tau$ above 16 km retrieved from BeCOOL and CALIOP. CALIOP L2 operational algorithm partially detects the thin cirrus (white box on **c**, red segment on **g**) after a 80 km horizontal averaging and reports a single optical depth value for this 80 km leg around coincidence time: at this resolution, a single point is detected as cloudy, CALIOP algorithm is missing most of the cloud. Manual retrievals of cloud optical depth using the same lidar ratio for both instrument (blue triangles on **f** and **g**) show an excellent agreement.

can fully be explained by the difference in assumed lidar ratio (48.9 *vs* 21.8). Furthermore, this example shows that the optical depth of thinnest clouds could be largely overestimated by BeCOOL retrieval due to an inaccurate lidar ratio choice. This key parameter can only be constrained for thicker clouds that significantly attenuate the lidar beam (optical depth $\gtrsim 10^{-2}$), but an *a priori* lidar ratio value has always to be assumed for thinnest features such as this cirrus. Yet, in most cases, we expect an
inaccurate choice of lidar ratio to only result in a factor $\sim 2$ difference in $\tau$ between CALIOP and our retrieval.

We can attempt to estimate the horizontal extension of this UTTC assuming an horizontal extension of a few hundreds of kilometers along both instrument's track: this cirrus could have an area greater than $10^5$ km$^2$, which is the order of magnitude observed by Peter et al. (2003) for UTTCs.

Regarding backscattered power, the contrast between the cloud and the surrounding clear sky is higher at 808 than 532 nm
(due to the strong wavelength dependency of Rayleigh scattering, emphasized by Peter et al. (2003)). This is why, in addition to



a lower absolute noise level, very thin features are more easily detected with BeCOOL. Such thin layers are sometimes clearer in the depolarization ratio (not shown here) and should have a stronger signature in CALIOP's second channel (1064 nm) but this channel is unfortunately too noisy, and the operational layer detection algorithm only relies for now on the 532 nm channel. Further reprocessing of CALIOP's observations are expected to improve the detection and retrieval of very thin clouds.

Vaillant de Guélis et al. (2021) recently introduced a new 2-dimensional multi-channel cloud detection algorithm for CALIOP. Preliminary tests on collocated BeCOOL/CALIOP's observations over very thin clouds show large improvements in cloud layer detection. Other projects of CALIOP reprocessing rely on machine learning techniques to detect optically thin clouds (Wang et al., 2019).

### 3.3 Third case study: convective clouds

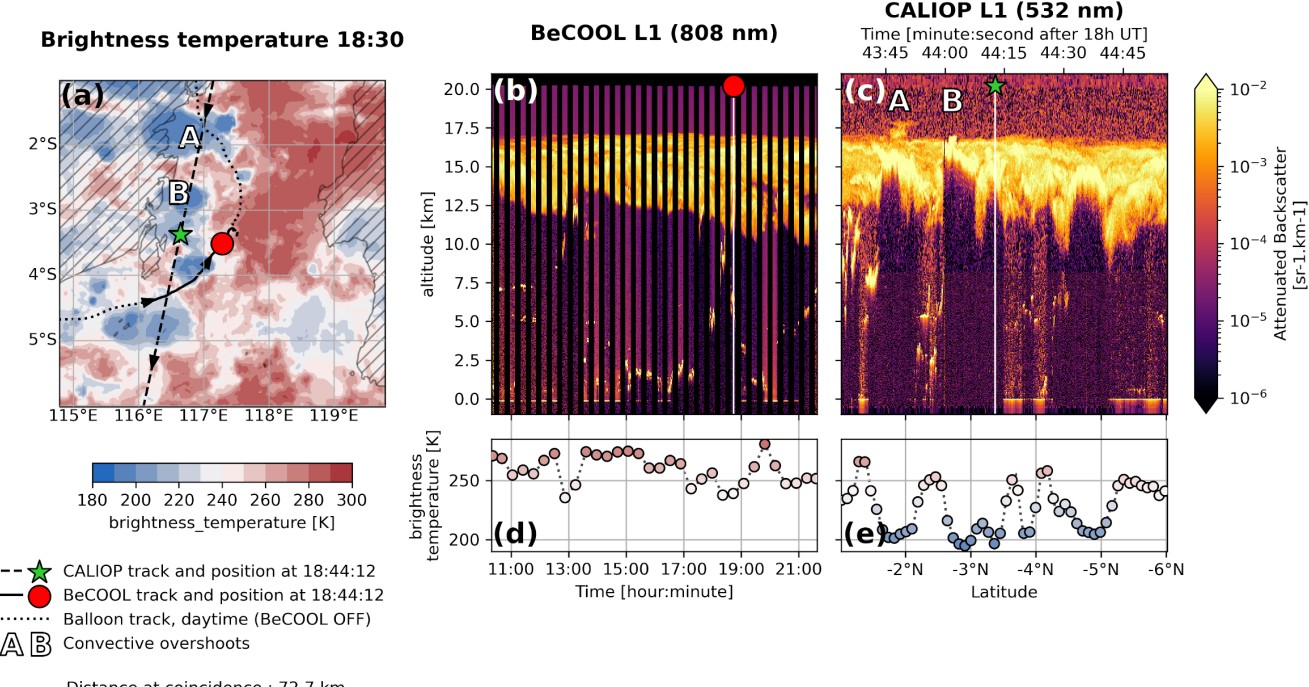

**Figure 4.** Third case study: convective cloud, 2021-11-27. **a**: 11 μm brightness temperature map at 18:30 UTC; **b**: BeCOOL L1 curtain (along the solid line on the map); **c**: CALIOP L1 curtain (along the dashed line on the map); **d**, **e**: time series of brightness temperature under the balloon and the satellite. White capital letters on **a** and **c** show two overshoots detected by CALIOP.

Figure 4 displays a CALIOP-BeCOOL coincidence which occured on 27 November 2021 off the southeast coast of Borneo island, Indonesia (∼4° S, 117° E; distance at coincidence: 72.7 km) for the third BeCOOL flight. CALIOP overflew several convective cells, capturing two events of convection overshooting the main cloud top (white capital letters on Fig. 4.a-c). The



first one (A) has an apparent diameter of 40 km and seems to be fading out at the time of overpass, having appeared 1 to 2 h before (see the hourly maps on Fig. A2). The second one (B) seems to be popping up right under the satellite and has an

apparent diameter of 15 km. The core of this cell is characterized by a very strong backscatter and a small penetration depth: the highest part of this cloud is very dense and optically thick. Those two structures clearly overshoot the extensive 17 km cloud top which appears on both curtains and is likely a high-altitude anvil. As shown by the brightness temperature maps, BeCOOL flew all night long around the convective cells, measuring the edges of anvils and revealing the evolution of the multilayered cloud structure. No clear sign of overshoot residual (i.e., cloud above the extensive 17 km deck) appears in BeCOOL data.

There could be different reasons for this apparent disagreement in overshoot detection. First, it is worth mentioning that "overshoots" would have different visual aspects in CALIOP and BeCOOL curtains. Assuming a wind difference of 7.5 $\mathrm{ms}^{-1}$ between the balloon and the cloud top (mean value over the Maritime Continent along the three flights, according to ERA5 reanalysis), an overshoot with a size of 40 km would be overflown for about 1.5 hours, a duration comparable to its lifetime (Dauhut et al., 2018; Lee et al., 2019). Thus, it is likely that overshoots in BeCOOL curtains will exhibit a different shape

(aspect ratio, ...) compared to CALIOP's, and cannot be identified as clearly. Over the whole campaign, we found no obvious observation in BeCOOL's data of an "overshoot" similar to the protrusion detected by CALIOP in this example. Since convective overshoots are one of the scientific targets of BeCOOL, we investigated further the probability of overflying very deep convection with the balloons using 11 $\mu$m Brightness Temperature (BT) data. We defined a $T_{overshoot} = 200$ K BT threshold as a proxy for potentially overshooting convection, based on the comparison between CALIOP and BT maps in this case

study. Over the Maritime continent and during the campaign, about 1 % of the pixels have BT lower than $T_{overshoot}$, whereas such cold pixels are 3 times less likely to occur along the balloon tracks (0.3 % of the observations over this area). Such low frequency of observation of "cold" cloud scenes constitutes a "warm" sampling bias for our 3 flights. Nevertheless, extending this analysis to all 17 Strateole-2 campaign balloons, we did not find conclusive evidence of a systematic sampling bias, which tends to discard the hypothesis that a dynamical effect (such as flow divergence) prevents the balloon from flying over over-

shooting tops. Targeting such relatively rare, sparse and small-scale structures with a limited instrumented fleet may require the use of steerable balloons.

## 4 Statistical description

### 4.1 Cloud coverage and scene complexity

A summary of BeCOOL profile classification over the Maritime Continent and Central Pacific Ocean is provided in Table 3.

A striking contrast appears between the two regions: over the Maritime Continent, convection is detected in up to 13 % of the profiles and clear-sky scenes are almost absent (0.2 %). On the contrary, more frequent clear sky profiles (12 %) and far less convective ones (0.6 %) are found over the Central Pacific. Over the Central Pacific, 71 % of the profiles present only cirrus, 60 % only TTL cirrus. Over the Maritime continent, half of the profiles correspond to a complex combination of different types of clouds, here reported as "Mixed multilayered scenes", while we only report 13 % of such profiles over the Central Pacific

Ocean. Although several types of scenes are gathered in this "mixed multilayered" class, a great part of them could be somehow




**Table 3.** Main profile classification, percentages of 10 minutes averaged profiles. Details on this classification can be found in Sect. 2.1.

|  | Full Area | Indian Ocean | Maritime Continent | Central Pacific Ocean |
|---|---|---|---|---|
| Longitude boundaries | 55 to 230° E | 55 to 95° E | 95 to 165° E | 165 to 230° E |
| Number of 10-min profiles | 4050 | 649 | 1091 | 2310 |
| Clear sky | 8 % | 6 % | 0.2 % | 12 % |
| Deep convection | 5 % | 7 % | 13 % | 1 % |
| Cirrus only *(TTL cirrus only)* | 57 % (43 %) | 40 % (15 %) | 36 % (21 %) | 71 % (60 %) |
| Mid-level cloud only | 3 % | 4 % | 0.6 % | 4 % |
| Mixed multilayered scene | 28 % | 43 % | 50 % | 13 % |

related to different stages of convective activity: developing convection, detrainment and/or precipitation. The campaign took place during La Niña phase of the El Niño Southern Oscillation (Southern Oscillation Index), the strong contrast in convective activity between the Maritime Continent and Central Pacific Ocean is typical of this ENSO phase (e.g. Gage and Reid, 1987).

**Table 4.** Frequency of occurrence of cirrus in BeCOOL profiles with different thresholds on optical depth, percentage of 10-minute profiles. Bold font stands for TTL cirrus.

|  | Full Area | Indian Ocean | Maritime Continent | Central Pacific Ocean |
|---|---|---|---|---|
| All cirrus | 87 % **59 %** | 83 % **39 %** | 94 % **47 %** | 84 % **71 %** |
| $\tau > 2 \cdot 10^{-3}$ | 75 % **43 %** | 71 % **16 %** | 92 % **39 %** | 68 % **53 %** |
| $\tau > 3 \cdot 10^{-2}$ | 59 % **25 %** | 64 % **7 %** | 84 % **28 %** | 46 % **28 %** |
| $\tau > 0.1$ | 48 % **15 %** | 57 % **5 %** | 77 % **21 %** | 32 % **15 %** |

Table 4 summarizes the occurrence of cirrus and TTL cirrus for several optical depth thresholds. Regardless of their optical depth, cirrus are detected in 87 % of all profiles with a weak regional contrast: from 83 % over Indian Ocean to 94 % over the Maritime Continent. The regional contrast is more pronounced for TTL cirrus, which are detected in 39 % of the profiles over the Maritime Continent and 71 % over the Central Pacific Ocean. The thresholds on optical depth show what would be detected by less sensitive instrument: CALIOP ($\tau > 0.002$), human bare eye (visible cirrus, $\tau > 0.03$), passive radiometers ($\tau > 0.1$). The cirrus cloud cover estimate strongly depends on this detection threshold: over the full area, not taking into account the thinnest cirrus clouds (optical depth below $2 \cdot 10^{-3}$) reduces the total cirrus coverage by 12%, and 16% for the TTL cirrus only. A passive radiometer insensitive to clouds with an optical depth below 0.1 (for example, onboard geostationary satellites) would only detect 1 cirrus out of 2, and 1 TTL cirrus out of 4. Thus, with BeCOOL's sensitivity, the estimated cirrus cover is significantly increased compared to what is derived from space-borne instruments.



## 4.2 Optical and geometrical properties of high-level clouds from BeCOOL and comparison with CALIOP

Statistics of cloud properties (optical depth, top and base altitude) have been compiled for all BeCOOL Level 2 profiles from the 2021 Strateole-2 campaign. They are compared with CALIOP nighttime profiles measured during the flight period of the microlidars, over the area covered by the balloons (from -10 to 5° N, 50 to 230° E, dashed white box on Fig. 1). All clouds with a reported base altitude below 5 km have been removed from both datasets to focus on free-tropospheric and TTL clouds. This includes deep convective clouds with full attenuation of the beam. Unlike the previous section, we use here the actual
lidar-determined top and base altitude instead of the modified altitudes used for the cloud classification.

Figure 5 shows histograms of optical depth for clouds detected above 5 km by the two instruments. An excellent agreement between the distributions appears from $2 \cdot 10^{-3}$ to $\sim 1$, the frequencies of occurrence both decrease as $\tau^{-1}$. For BeCOOL, this power law is valid down to $\tau \simeq 10^{-4}$ where the distribution reaches a maximum, whereas a clear cut-off appears at a larger optical depth $\tau \simeq 2 \cdot 10^{-3}$ in CALIOP's distribution, below which the cloud frequency sharply decreases. 20% of the cloud
layers detected by BeCOOL have an optical depth below $2 \cdot 10^{-3}$. They appear in 26 % of the profiles. For CALIOP, such ultrathin clouds only account for 0.5 % of all detected clouds and are reported in 1 % of the profiles.

Figure 6 shows the distributions of cloud top altitude, base altitude and geometrical depth for all clouds, and separately for clouds with an optical depth larger or smaller than a $2 \cdot 10^{-3}$ threshold. For all clouds, BeCOOL's top altitude distribution shows a sharp mode peaking between 17 and 17.5 km, and a wider base altitude distribution peaking between 16.5 and 17 km.
The mean geometrical depth is 1.9 km. Considering only the clouds with an optical depth larger than $2 \cdot 10^{-3}$, the top altitude distribution remains almost unchanged, with a slightly smoother mode just below, between 16.5 and 17 km, base altitude also shows a smooth mode between 15 and 16.5 km, the mean geometrical depth is 2.25 km. Considering only cirrus with an optical depth below $2 \cdot 10^{-3}$, both top and base altitude distributions shows a very sharp mode peaking between 17.5 and 18 km for the top and between 17 and 17.5 km. Correspondingly, the mean geometrical depth is 480 m. Almost 75 % of those clouds
lie within the TTL, and about 50 % have their base above 16.5 km. Hence, not only does BeCOOL perform well in detecting ultrathin TTL clouds (an expected result considering its high SNR in near-field), more importantly, such clouds are detected in almost 20 % of the profiles. Now comparing BeCOOL to CALIOP, a very similar total top altitude distribution is seen, yet with a peak shifted towards lower altitudes, between 16.5 and 17 km. The base altitude distribution of CALIOP does not exhibit any sharp mode and does not extend as high as BeCOOL's, it appears quite uniform between 10.5 and 15.5 km and
decreases below, the mean geometrical depth is 2.1 km. Those distributions remain unchanged when considering only clouds with an optical depth larger than $2 \cdot 10^{-3}$ as they account for 99.5 % of all clouds. The agreement with BeCOOL's top altitude distribution is almost perfect for those clouds, base altitude distributions are in better agreement, although CALIOP's still does not extend as high as BECOOL's. The mean geometrical depth of those clouds is still 2.1 km, which is slightly lower than for BeCOOL, but the distribution does not extend as much to small depths. For CALIOP, according to Yorks et al. (2011),
multiple scattering effects due to the large size of the lidar footprint tend to lower the apparent cloud base altitude, enhancing the apparent geometrical depth of the cloud. This explains the differences between base altitude distributions and geometrical depth, while top altitude distributions show an excellent agreement. In striking contrast with BeCOOL, clouds with an optical





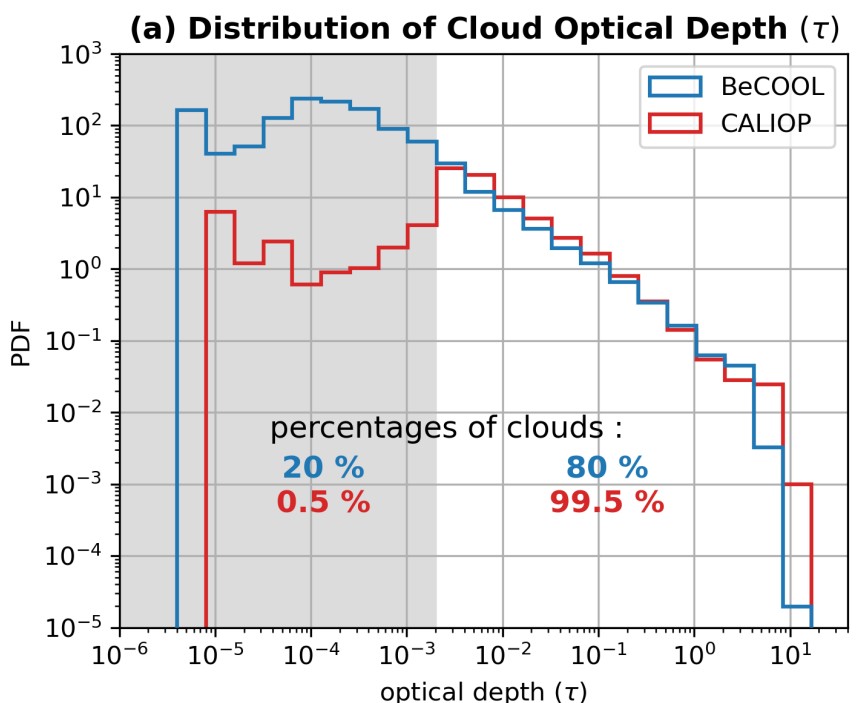

**Figure 5.** Statistical comparison of cloud layers properties detected by BeCOOL and CALIOP. **a**: Probability Density Functions of optical depth of all clouds above 5 km, the grey shading highlights the low optical depth, up to $2 \cdot 10^{-3}$, where the distributions diverge. The percentages of detected clouds with an optical depth lower/greater than this $2 \cdot 10^{-3}$ threshold are reported on the figure. **b**: Percentages of lidar profiles showing clouds with an optical depth lower/greater than the $2 \cdot 10^{-3}$ threshold. Both types of clouds can appear in a single lidar profile.

depth below $2 \cdot 10^{-3}$ only represent 0.5 % of CALIOP's cloud database, and their top/base altitude distribution is wide and does not show any pronounced mode.

The excellent agreement between the distributions for optical depths larger than $2 \cdot 10^{-3}$ shows that, despite their limited sampling, balloon-borne observations are representative of the area studied. On the contrary, for small optical depths, the comparison highlights BeCOOL's unique ability to detect ultrathin TTL cirrus. As shown in Sect. 3.2, such cirrus can persist throughout the night below the balloon, and appear homogeneous. They usually lay either right underneath the cold



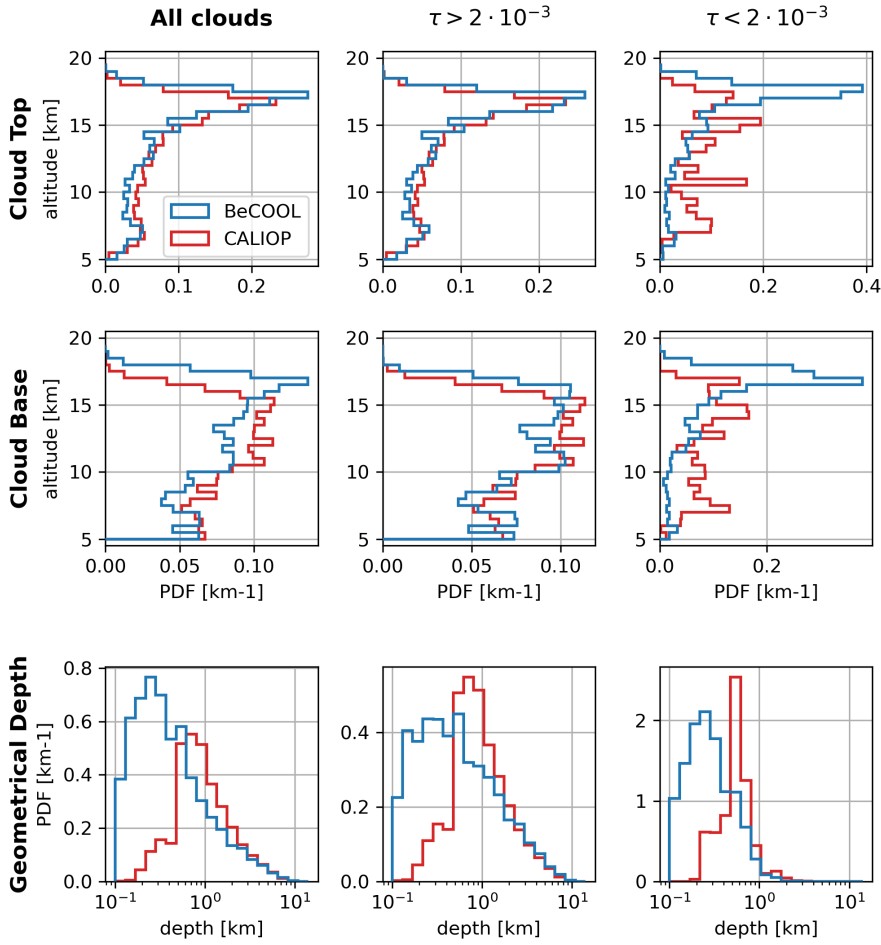

**Figure 6.** Statistical comparison of cloud layers properties detected by BeCOOL and CALIOP. Probability Density Functions of (lines) top altitude, base altitude, and geometrical depth for (columns) all clouds, clouds with an optical depth above/below the $2 \cdot 10^{-3}$ threshold.

point tropopause, or a local temperature minimum, according to collocated temperature profiles from GPS-RO soundings (not
shown). These characteristics make those thin cirrus similar to UTTCs defined by Peter et al. (2003) and Luo et al. (2003) from
the airborne measurements.

Mean top altitude $\overline{z}_{\mathbf{top}}$ and geometrical depth $\overline{\mathbf{\Delta z}}$ of TTL cirrus for different ranges of optical depth $\tau$ are summarized in
Table 5. The mean top altitude is fairly constant regardless of the value of $\tau$, whereas, as expected, the geometrical depth is
clearly correlated with $\tau$. The depth of the cloud layer is often used as a free parameter for Lagrangian parcel box models
of cirrus and stratospheric dehydration (e.g., Fueglistaler and Baker, 2006; Spichtinger and Krämer, 2013; Schoeberl et al.,
2014; Poshyvailo et al., 2018; Nützel et al., 2019). Here, BeCOOL observations suggest typical depths of TTL cirrus ranging



**Table 5.** Mean top altitude $\overline{z}_{top}$ and geometrical depth $\overline{\Delta z}$ of TTL cirrus for different ranges of optical depth $\tau$.

|  | % of TTL cirrus | $\overline{z}_{top}$ | $\overline{\Delta z}$ |
|---|---|---|---|
| All TTL cirrus | 100 % | 16.9 km | 950 m |
| $\tau < 2 \cdot 10^{-3}$ | 39 % | 17.1 km | 430 m |
| $2 \cdot 10^{-3} < \tau < 3 \cdot 10^{-2}$ | 33 % | 16.8 km | 740 m |
| $3 \cdot 10^{-2} < \tau < 10^{-1}$ | 15 % | 16.8 km | 1360 m |
| $10^{-1} < \tau$ | 13 % | 17.1 km | 2130 m |

from less than 0.5 km (optically thinner ones) to about 2 km (optically thicker ones), with a mean of $\sim$1 km, which is overall compatible with the values used in modeling studies.

### 4.3 Cirrus and temperature anomalies

This common detection of very thin TTL cirrus layers by BeCOOL raises the question of the processes responsible for their formation. Following recent papers (Kim et al., 2016; Podglajen et al., 2018; Chang and L'Ecuyer, 2020; Bramberger et al., 2022), we investigated the relationship between TTL clouds and temperature anomalies. Those previous studies highlighted the ubiquitous influence of wave-induced temperature anomalies $T'$ and their vertical gradient $\mathrm{d}T'/\mathrm{d}z$ on cirrus clouds. Following Chang and L'Ecuyer (2020), temperature anomalies have been computed using GPS-Radio Occultation (GPS-RO) tempera-

ture profiles from the Constellation Observing System for Meteorology, Ionosphere, and Climate (COSMIC) Data Analysis and Archive Center (CDAAC) of the University Corporation for Atmospheric Research (UCAR). First, for each BeCOOL flight and each night, a background temperature profile has been determined by averaging all GPS-RO profiles within in a 5° latitude × 10° longitude box centered on the balloon mean position over a 14-day rolling window. Then, for each night, all GPS-RO profiles falling within a 300 km-radius of the balloon and between 3 h before the first lidar observation and 3 h

after the last one were selected. Finally, for each lidar observation, the corresponding temperature anomaly profile was computed as the difference between the closest GPS-RO profile in time among the selected ones and the background. We then split the cloudy lidar data points within the TTL into four categories depending on the temperature anomaly, corresponding to wave phases with positive or negative temperature anomaly $T'$ and lapse-rate anomaly $\mathrm{d}T'/\mathrm{d}z$. Figure 7 shows the results for the whole campaign and 4 different 1 km ranges, in a similar fashion as Figure 3 of Chang and L'Ecuyer (2020). Our

results are overall consistent with that previous study, placing almost half of the clouds in the wave phase in which both $T'$ and $\mathrm{d}T'/\mathrm{d}z$ are negative. As explained in Kim et al. (2016), assuming that temperature anomalies are induced by gravity waves with a downward propagating phase, negative anomalies of $\mathrm{d}T'/\mathrm{d}z$ correspond to positive vertical wind anomalies, thus to cooling conditions that lower the condensation point. Hence, our observations also suggest favorable conditions for TTL cirrus presence in the cold and cooling phase of gravity waves, which might be related to the influence of the wave-induced satu-

ration anomalies on the formation of the ice crystals (Kim et al., 2016) and/or on their subsequent growth and sedimentation Podglajen et al. (2018).



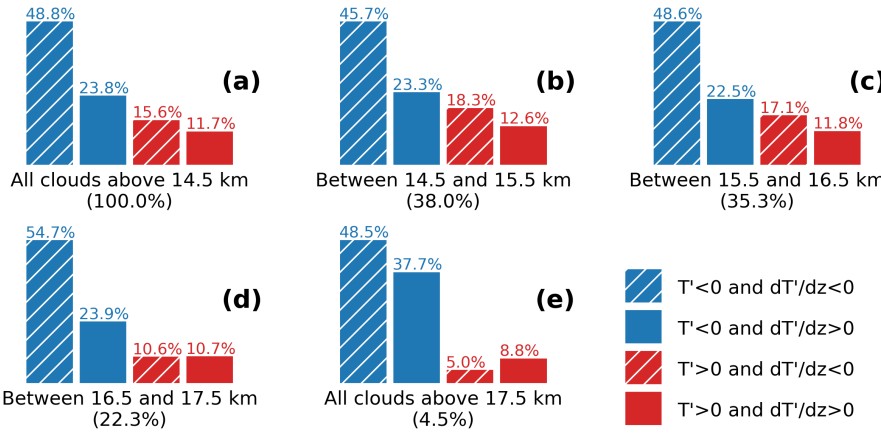

**Figure 7.** Fraction of cloudy BeCOOL lidar bins within 4 wave phases. **a**: cloud population for the whole TTL; **b**-**e**: population in four different 1 km vertical layers. The percentage in parentheses is the portion of clouds in that layer relative to all TTL clouds.

## 5 Conclusions

Three BeCOOL microlidars were flown during the Strateole-2 scientific campaign in the boreal winter 2021-2022. They provide the first long-duration balloon-borne cloud lidar dataset, covering the equatorial region from the Indian Ocean up to the
middle of the Pacific Ocean. These observations were compared with space-borne lidar observations from CALIOP.

Case studies of collocated BeCOOL/CALIOP observations for two different types of cirrus clouds demonstrated both the agreement between the two lidars for thicker clouds and BeCOOL's enhanced sensitivity to tenuous clouds. A longer integration time and the proximity of BeCOOL to the studied clouds are responsible for its higher sensitivity. A third case study over convective anvils illustrated the low likelihood of observing short-lived, small-scale structures, such as overshooting convective
cloud tops, within a limited dataset gathered from freely drifting balloons. Targeting specific uncommon cloud features would require the use of steerable balloons.

Occurrence statistics of different cloud types and profile classification reveal that cirrus clouds are ubiquitous over the area overflown by the balloons, with a wide range of optical depth covering several orders of magnitude. Cirrus clouds are detected in 87 % of the lidar profiles, with a limited regional variability during the campaign. On the contrary, the deep convective cloud
cover varies very significantly between the studied regions, ranging from 13 % of the observations over the Maritime Continent to less than 1 % over the Central Pacific Ocean. TTL cirrus, i.e. cirrus with a could base above 14 km, are found in 59 % of all profiles (and 71 % over the Central Pacific Ocean). Their mean top altitude is 17 km, and does not depend on their optical depth. Their geometrical depth ranges from less than 0.1 to 4 km, with an overall mean of ∼1 km. Ultrathin TTL cirrus, with optical depth below the detection threshold of CALIOP ($\tau < 2 \cdot 10^{-3}$), are reported in 16 % of the lidar profiles, and have a
mean geometrical depth of about 400 m. Those very thin TTL clouds are reminiscent of Ultrathin Tropical Tropopause Clouds described by Peter et al. (2003), in particular with respect to their small vertical extension, huge horizontal extension and lateral





homogeneity. These clouds play a significant role in the dehydration process of air masses entering the stratosphere (e.g., Jensen et al., 1996; Schoeberl et al., 2019). An ongoing study investigates their radiative impact from BeCOOL's measurements. Our observations also confirm the ubiquitous relationship between waves and tropical cirrus clouds found in previous papers (Kim
et al., 2016; Chang and L'Ecuyer, 2020; Bramberger et al., 2022): TTL cirrus are more common in the cold and cooling phase of waves. Future work will focus on characterizing the horizontal scales and lifetimes of TTL cirrus combining CALIOP and BeCOOL, in order to elucidate the link between waves and TTL cirrus life cycle.

*Data availability.* The Strateole-2 BeCOOL data set is available at https://data.ipsl.fr/catalog/strateole2/. CALIOP data was downloaded from AERIS/ICARE datacenter (https://www.icare.univ-lille.fr/calipso/products/). GPS-RO data can be accessed at COSMIC Data Anal-
ysis and Archive Center (https://cdaac-www.cosmic.ucar.edu/cdaac/index.html). The merged IR satellite images were collected from the NOAA/NCEP GPM_MERGIR product, available at https://disc.gsfc.nasa.gov/datasets/GPM_MERGIR_1/summary.

**Appendix A: Additional brightness temperature maps for the case studies**

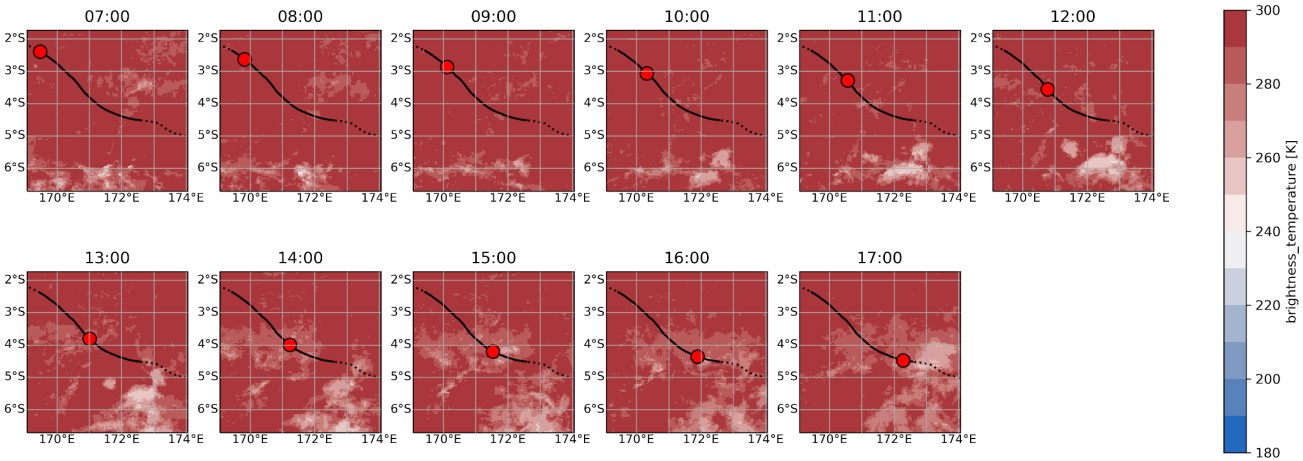

**Figure A1.** Hourly brightness temperature maps for case study 1, 2021-11-29. Red dot is the balloon position, solid (dotted) line is the nighttime (daytime) balloon track.

*Author contributions.* TL, FR and AP conceived the study. TL performed the study with scientific support from FR, AP and JP. TL wrote the paper with contributions from FR and AP. VM designed and built BeCOOL microlidar. All authors agreed on the final version.



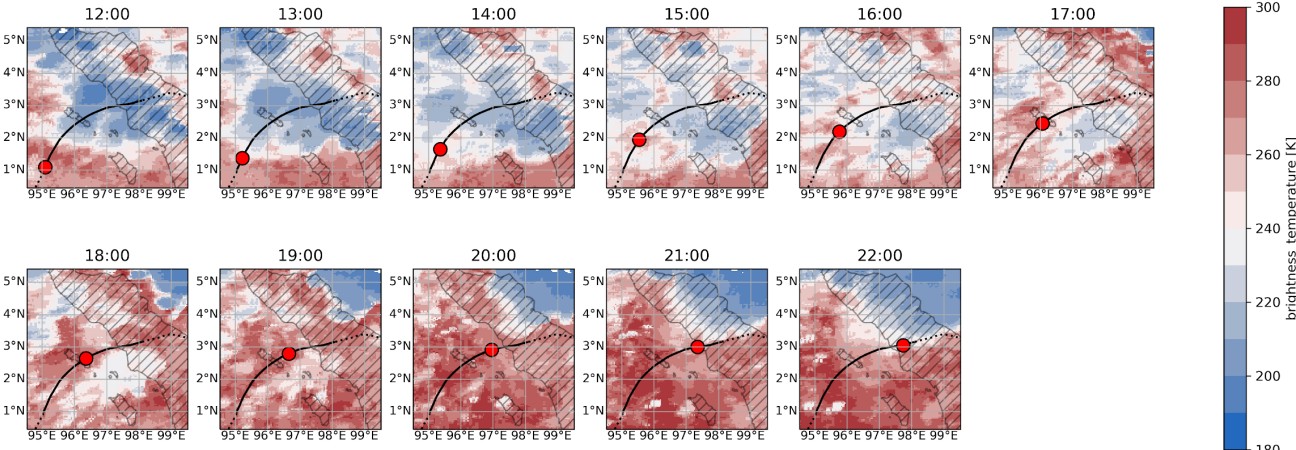

**Figure A2.** Hourly brightness temperature maps or case study 2, 2021-11-27. Red dot is the balloon position, solid (dotted) line is the nighttime (daytime) balloon track.

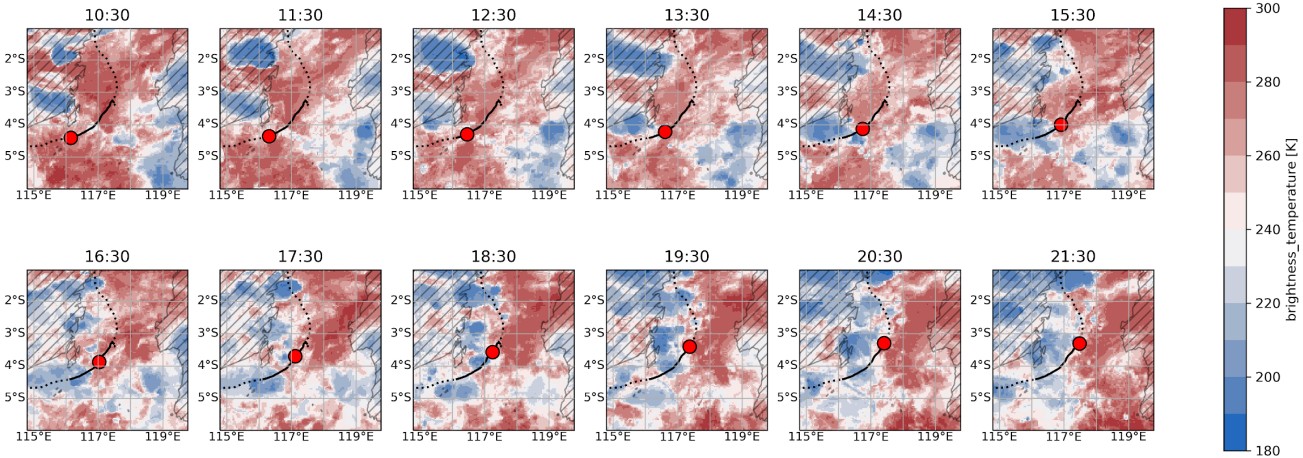

**Figure A3.** Hourly brightness temperature maps for case study 2, 2021-11-27. Red dot is the balloon position, solid (dotted) line is the nighttime (daytime) balloon track.

*Competing interests.* At least one of the (co-)authors is a member of the editorial board of Atmospheric Chemistry and Physics.

*Acknowledgements.* The balloon-borne BeCOOL data were collected as part of Strateole-2, which is sponsored by CNES, CNRS/ INSU, NSF, and ESA. TL acknowledges support by a doctoral grant from France's Centre National d'Études Spatiales (CNES). The authors would



like to thank the numerous developers that contributed to the free and open-source tools used for the data analysis and visualization, in particular xarray (Hoyer and Hamman, 2017, DOI: https://doi.org/10.5334/jors.148) and Matplotlib (Hunter, 2007, DOI: 10.1109/MCSE.2007.55).



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
