# Peer review of "Extensive coverage of ultrathin Tropical Tropopause Layer cirrus clouds revealed by balloon-borne lidar observations"

_EGUsphere, 2023_

## Referee Comment (RC1)

This paper compares the nighttime observations of mid- and high-level clouds from balloon-borne microlidars BeCOOL obtained from October 2021 to January 2022 near the equator from Indian to Central Pacific Oceans with the observations collected by the space-borne lidar CALIOP. Three collocated (in space and time) case studies are first presented followed by a statistical comparison of the cloud optical and geometrical properties from all observations performed during the campaign with those obtained by CALIOP over the same region during the flight time of the balloons. A last section analyze the link between Tropical Tropopause Layer (TTL) clouds and temperature anomalies provided by GPS-Radio Occultation.

BeCOOL is a system designed for TTL cirrus and convective overshoot monitoring. The study demonstrates the very good agreement between BeCOOL and CALIOP and the superiority of BeCOOL to detect ultrathin clouds, validating its usefulness for the study of TTL cirrus. However, it also suggests that obtaining BeCOOL observations of overshooting convective cloud tops would require steerable balloons as their occurrences are rare and the sampling during such campaign too small.

The article is well-organized, easy to read and clear. The figures perfectly illustrate the main results and are of the highest quality. I would recommend its publication after taking into account the comments below.

My only main concern is that the descriptions of the processing of Level 1 and Level 2 products are not yet available, making the reviewing somewhat difficult as we are not informed by potential bias and errors in the cloud detection and optical properties retrieval methods. It would be great that the authors provide information on the detection method and the optical depth retrieval.

Below is a list of minor comments.

Line 13: "T̶tropics"

Lines 81–83: "For the cloud classification, in order to focus on the main part of clouds, top and base altitudes are slightly modified so that 15 % of the cloud optical depth lay above the new top altitude, and 15 % below the new base altitude." – What is the rationale for doing this? I understand it decreases cloud top altitude and increases cloud base altitude. How this relate with the results from Sect. 4.2 where such manipulation is not performed?

Figs. 2g and 4e: Use "° S" instead of "-°N" for consistency with other plots and text.

Line 143: "along the 570-km track"

Lines 189–191: "Regarding backscattered power, the contrast between the cloud and the surrounding clear sky is higher at 808 than 532 nm (due to the strong wavelength dependency of Rayleigh scattering, emphasized by Peter et al. (2003)). This is why, in addition to a lower absolute noise level, very thin features are more easily detected with BeCOOL." – In what proportion play the wavelength and the noise?

Line 211: "7.5 m·s⁻¹" → Line 211: "7.5 m$\cdot$s$^{-1}$"

Line 220: "Over the Maritime Continent and during the campaign, about 1 % of the pixels of the BT maps have values lower than" – Are you considering the collocated 5°×5° BT maps only or the BT maps of the whole Maritime Continent region?

Line 223: "all 17 Strateole-2 campaign balloons" – Please explained here, in the Introduction, or in Sect. 2.1, that this Strateole-2 campaign releases many balloons whose three with a BeCOOL lidar (if this is correct).

Table 3: Mention in the caption this is for BeCOOL observations.

Line 232: "(0.6 %)" – 1 % in Table 3. 0.6 % is for the Mid-level cloud only over Maritime Continent. Keep same decimal notation (at least for value < 1 %) in the Table.

Line 233: "Maritime Continent"

Lines 241–242: "which are detected in 39 % of the profiles over the Maritime Continent" – Table 3: Maritime Continent = 47 %; Indian Ocean = 39 %

Table 4: "Frequency of occurrence of cirrus (cloud base > 10 km) in BeCOOL profiles with different thresholds on optical depth, percentage of 10-minute profiles. Bold font stands for TTL cirrus (cloud base > 14 km)."

Lines 242–243: "The thresholds on optical depth show what would be detected by less sensitive instrument: CALIOP (τ > 0.002), human bare eye (visible cirrus, τ > 0.03), passive radiometers (τ > 0.1)." – It would be interesting to add figures (in Appendix?) of the BeCOOL mask color-coded with those OD thresholds corresponding to Figs. 1, 2b, 3b, and 4b. This would allow to better see the BeCOOL's enhanced detection capability.

Line 249: "Optical and geometrical properties of mid- and high-level clouds"

Lines 250–251: "for all BeCOOL Level 2 profiles from the 2021 Strateole-2 campaign" – Does it represent more data than the 3 flights shown in previous sections?

Lines 252–253: "All clouds with a reported base altitude below 5 km have been removed from both datasets to focus on free-tropospheric and TTL clouds." – Why this is not directly a condition for the cloud classification (Sect. 2.1)?

Lines 254–255: "Unlike the previous section, we use here the actual lidar-determined top and base altitude instead of the modified altitudes used for the cloud classification." – I don't understand the rationale for this. See comment for Lines 81–83.

Fig. 5b: It shows that 82 % of the lidar profiles show clouds with OD > 2·10$^{-3}$. In Table 4, it was 75 %. What explains that the percentage is now higher? I might have understood if the value was lower since you removed mid-level clouds with base below 5 km from datasets (for some reasons I don't understand). Could it be due to the fact you don't use the same definition for cloud top and base altitudes in both sections (for some reasons I don't understand)?

Line 269: "and between 17 and 17.5 km for the base."

Lines 279–282: "For CALIOP, according to Yorks et al. (2011), multiple scattering effects due to the large size of the lidar footprint tend to lower the apparent cloud base altitude, enhancing the apparent geometrical depth of the cloud. This explains the differences between base altitude distributions and geometrical depth, while top altitude distributions show an excellent agreement." – Actually, this effect is mainly due to a non-ideal transient response of the PMTs (Lu et al. 2013, 2020). The effect is more visible in liquid clouds. I suspect the "Geometrically thin (a few hundred meters), horizontally extensive mid-level clouds are often found above, below 10 km, mainly between 5 and 8 km; they typically have large backscatter and are likely pure liquid or mixed-phase clouds." (Lines 68-70), to play a significant role in the geometrical depth difference you observed. Another reason for this effect comes from the difference in horizontal averaging in the detection algorithms. Compare to the detections performed at ~5-km horizontal resolution in the BeCOOL data, the detections performed at 20- and 80-km horizontal resolution in the CALIOP data can introduce a low bias in the cirrus cloud base retrieval when the cloud base altitude fluctuates at smaller scales. This does not affect the cirrus cloud tops very much are they appear flatter at those scales.
- Lu, X., Hu, Y., Liu, Z., Zeng, S., and Trepte, C.: CALIOP receiver transient response study, SPIE Optical Engineering + Applications, San Diego, California, United States, https://doi.org/10.1117/12.2033589, 2013.
- Lu, X., Hu, Y., Vaughan, M., Rodier, S., Trepte, C., Lucker, P., and Omar, A.: New attenuated backscatter profile by removing the CALIOP receiver's transient response, J. Quant. Spectrosc. Radiat. Transfer, 255, 107244, https://doi.org/10.1016/j.jqsrt.2020.107244, 2020.

Line 289: "GPS-RO" – Define here (where first mentioned).

Table 5: Mention in the caption this is for BeCOOL observations.

Line 336: "couldcloud base"

Lines 338–339: "Ultrathin TTL cirrus, with optical depth below the detection threshold of CALIOP ($\tau < 2\cdot10^{-3}$ ), are reported in 16 % of the lidar profiles" – Is 16 % coming from Table 4? My understanding is that this value corresponds to profiles containing ultrathin TTL cirrus only. In Sect. 4.2, you mentioned that clouds with $\tau < 2\cdot10^{-3}$ appear in 26 % of the profiles, but this is not limited to TTL cirrus. I would guess that the ultrathin TTL cirrus are detected in the lidar profiles with a fraction between 16 % and 26 %.

Fig. A2: "for case study 2"

---

## Referee Comment (RC2)

Review of

**Observations of Tropical Tropopause Layer clouds**
**from a balloon-borne lidar**

by Lesigne et al.

**General:**  This study reports on  measurements of cirrus clouds using  the new balloon-borne microlidar  BeCOOL, operated during three flights onboard superpressure balloons as part of the Strateole-2 measurement campaign.  From  three collocated measurements of cirrus with different microphysical properties, BeCOOL was compared to CALIOP.  The agreement between the instruments was  very good and, moreover, BeCOOL was found  to be significantly more sensitive to thin cirrus compared to CALIOP.  A significant finding of the study is that a comparison of the frequencies of occurrence of cirrus with various optical depths reveals that CALIOP misses  ~20% of the cirrus, all within the range of $\tau < 2 \cdot 10^{-3}$.  Furthermore, all BeCOOL cirrus observations are statistically analyzed for different cirrus types in different regions and, in addition,  TTL cirrus top heights and  thicknesses,  classified according to optical depth.

This is an excellent and exciting study, presenting new  insights in the properties and distribution of high altitude tropical  cirrus clouds, based on high quality observations from a new instrument.
The manuscript is well organized, fluently written and the figures are appropriate. It was a pleasure to read and review this article.

I have only a few minor comments, which are listed in the specific comments, that I would recommend  to consider before publishing.
There is, however, one point on which the authors might have a second thought. To my feeling,  the study sells itself a bit short -  this is outlined in more detail in the specific comments to the abstract and title.

**Specific comments:**

**Abstract**: - I would include  a sentence on the goal of the study after the first sentence,
    see ACP guidelines for authors:
    https://www.atmospheric-chemistry-and-physics.net/policies/guidelines_for_authors.html

 - to my opinion it would be important to mention here that ~20% of the cirrus
    with $\tau < 2 \cdot 10^{-3}$ (cloud depth < 400m, cloud altitude > ~16km),
    which are mostly TTL cirrus, are not detected  by CALIOP.
    Would it  be going too far to conclude that cirrus radiative-climate feedback estimates
    may therefore need to be reconsidered?

    You have room to extend the abstrcat, it currently has 176 words and can be  up to  250 words-

**Title** :  Based on your exciting findings  (and looking into the ACP guidelines for the title), you might think about changing the title, e.g. to something like:

‚Observations of an unexpectedly/surprisingly high portion of Tropical Tropopause Layer clouds from a balloon-borne lidar‘

**Line 32ff**:   For space-borne lidar observations, it might be worth to cite Sourdeval et al. (2018).

**Line 36ff**:   For airborne measurements of cirrus including TTL observations, Krämer et al. (2020) could be added.

**Line 56**: ‚fiels campaign' → field  campaign

**Line 71:**  ‚The clouds' vertical structure can be fully resolved up to an optical depth $\tau_{max} \simeq 3, \dots$ '
          Later, in Figure 10,  optical depth  up to 10 are shown ?

**Line 173** ‚for such case' →  for such a case

**Line 315**   ‚… placing almost half of the clouds in the wave phase in which both $T'$ and $dT'/dz$ are negative.'

If I understand it right, these are the conditions of cirrus formation and the other cases represent aged cirrus, where  only the longer living cirrus are found, or ?  If this is true,  it could be mentioned in the discussion of this result in the following paragraph.

**Table 3**, caption**:** ‚BeCOOL main profile classification, percentages of 10 minutes averaged profiles. Details on this classification can be found in Sect. 2.1.'

**References:**

Sourdeval, O., Gryspeerdt, E., Krämer, M., Goren, T., Delanoë, J., Afchine, A., Hemmer, F., and Quaas, J.: Ice crystal number concentration estimates from lidar–radar satellite remote sensing – Part 1: Method and evaluation, Atmos. Chem. Phys., 18, 14327–14350, https://doi.org/10.5194/acp-18-14327-2018, 2018.

Krämer, M., Rolf, C., Spelten, N., Afchine, A., Fahey, D., Jensen, E., Khaykin, S., Kuhn, T., Lawson, P., Lykov, A., Pan, L. L., Riese, M., Rollins, A., Stroh, F., Thornberry, T., Wolf, V., Woods, S., Spichtinger, P., Quaas, J., and Sourdeval, O.: A microphysics guide to cirrus – Part 2: Climatologies of clouds and humidity from observations, Atmos. Chem. Phys., 20, 12569–12608, https://doi.org/10.5194/acp-20-12569-2020, 2020.

---

## Author Comment (AC1)

**Observations of TTL clouds from a balloon-borne lidar**
**Answer to reviewers**

We would like first to thank the three reviewers for their careful reading, relevant questions and comments that helped us improve the paper.

Finalizing the technical article describing BeCOOL instrument and data processing, we identified several issues which led us to reprocess all the observations. The main improvements are related to the convergence of constrained optical depth retrieval and the multiple scattering correction. We are confident that the accuracy of the processing has been improved. This reprocessing does not bring significant changes to the results of the paper but there is now a better agreement between BeCOOL's and CALIOP's retrieved optical depths, as illustrated by the two first case studies.

Replies to all referee comments are contained in this document. Referee comments are in black, or green for minor corrections. Our responses are in blue, line numbers of modifications (in the revised manuscript) in bold.

**RC1**
This paper compares the nighttime observations of mid- and high-level clouds from balloon- borne microlidars BeCOOL obtained from October 2021 to January 2022 near the equator from Indian to Central Pacific Oceans with the observations collected by the space-borne lidar CALIOP. Three collocated (in space and time) case studies are first presented followed by a statistical comparison of the cloud optical and geometrical properties from all observations performed during the campaign with those obtained by CALIOP over the same region during the flight time of the balloons. A last section analyze the link between Tropical Tropopause Layer (TTL) clouds and temperature anomalies provided by GPS-Radio Occultation. BeCOOL is a system designed for TTL cirrus and convective overshoot monitoring. The study demonstrates the very good agreement between BeCOOL and CALIOP and the superiority of BeCOOL to detect ultrathin clouds, validating its usefulness for the study of TTL cirrus.
However, it also suggests that obtaining BeCOOL observations of overshooting convective cloud tops would require steerable balloons as their occurrences are rare and the sampling during such campaign too small.

The article is well-organized, easy to read and clear. The figures perfectly illustrate the main results and are of the highest quality. I would recommend its publication after taking into account the comments below.

My only main concern is that the descriptions of the processing of Level 1 and Level 2 products are not yet available, making the reviewing somewhat difficult as we are not informed by potential bias and errors in the cloud detection and optical properties retrieval methods. It would be great that the authors provide information on the detection method and the optical depth retrieval.
We are aware that it was missing. One paragraph has been added to Sect. 2.1 to describe the optical depth retrieval. A detailed description of the instrument and the processing is the subject of an other paper that is currently finalized and will be submitted to AMT. (Lines **63-83**)

Below is a list of minor comments.

Line 13: "Ttropics"

Lines 81–83: "For the cloud classification, in order to focus on the main part of clouds, top and base altitudes are slightly modified so that 15 % of the cloud optical depth lay above the new top altitude, and 15 % below the new base altitude." – What is the rationale for doing this? I understand it decreases cloud top altitude and increases cloud base altitude. How this relate with the results from Sect. 4.2 where such manipulation is not performed?

This was an attempt to improve the cloud and scene classification, reducing the amount of unclassified clouds. The idea was to focus on the bulk of the cloud rather than on the potential secondary peaks on top or at the bottom of it. As the improvement was not that significant, and as the classification is not at the core of the study, we stepped back and kept a unique definition of cloud top and base altitudes.

Cloud top (base) altitude is set where the attenuated scattering ratio (ASR) is greater than the ASR at a reference clear-sky level just above (below) the cloud plus 5 times its standard deviation on a 10 points window around this reference level. The statistics presented in Tables 3 to 5 have changed since we kept this unique definition for cloud top and base altitudes. In particular the reported cirrus and TTL cirrus coverage are now lower since clouds are categorized as such from a single threshold on their base altitude. The former "modified" base altitude was higher than the unique base altitude defined here and tended to increase the number of clouds classified as "cirrus", or "TTL cirrus".

Figs. 2g and 4e: Use "° S" instead of "-°N" for consistency with other plots and text.

Line 143: "along the 570-km track"

Lines 189–191: "Regarding backscattered power, the contrast between the cloud and the surrounding clear sky is higher at 808 than 532 nm (due to the strong wavelength dependency of Rayleigh scattering, emphasized by Peter et al. (2003)). This is why, in addition to a lower absolute noise level, very thin features are more easily detected with BeCOOL." – In what proportion play the wavelength and the noise?

We will answer this question relying on the thin cirrus observation from the second case study, which is about the detection lower limit for CALIOP. The following figure shows the attenuated scattering ratios (ASR) for both instruments around the coincidence.

**Second case study : lidar profiles**

[Figure]

BeCOOL's ASR peaks at 15 while CALIOP's peaks below 5. The contrast between this cloud and the surrounding molecular atmosphere due to the spectral dependency of Rayleigh scattering is more than 3 times larger for BeCOOL than for CALIOP. Moreover, defining the noise level as the standard deviation of the ASR in clear sky just above the cloud (here in a 1 km-thick layer from 18 to 19 km), BeCOOL's noise level is 0.017, which is about 25 times smaller than CALIOP's noise level of 0.44. Eventually, defining the effective SNR as the ratio of the ASR peak within the cloud (-1 to account for the molecular contribution) and the noise level, BeCOOL's effective SNR is ~800 while CALIOP's is only 10.

This example shows that both the operating wavelength and the noise level contribute to enhance BeCOOL's capacity to detect thin cirrus, the main effect being the low noise level.

Line 211: "7.5 m·s -1 "

Line 220: "Over the Maritime cContinent and during the campaign, about 1 % of the pixels of the BT maps have BTvalues lower than" – Are you considering the collocated 5°×5° BT maps only or the BT maps of the whole Maritime Continent region?
We are considering here the BT maps of the whole Maritime Continent region.

Line 223: "all 17 Strateole-2 campaign balloons" – Please explained here, in the Introduction, or in Sect. 2.1, that this Strateole-2 campaign releases many balloons whose three with a BeCOOL lidar (if this is correct).
Indeed this is correct, this information was added in Sect. 2.1. (Line **59**)

Table 3: Mention in the caption this is for BeCOOL observations.

Line 232: "(0.6 %)" – 1 % in Table 3. 0.6 % is for the Mid-level cloud only over Maritime Continent. Keep same decimal notation (at least for value < 1 %) in the Table.

Line 233: "Maritime cContinent"

Lines 241–242: "which are detected in 39 % of the profiles over the Maritime Continent" – Table 3: Maritime Continent = 47 %; Indian Ocean = 39 %

Table 4: "Frequency of occurrence of cirrus (cloud base > 10 km) in BeCOOL profiles with different thresholds on optical depth, percentage of 10-minute profiles. Bold font stands for TTL cirrus (cloud base > 14 km)."

Lines 242–243: "The thresholds on optical depth show what would be detected by less sensitive instrument: CALIOP (τ > 0.002), human bare eye (visible cirrus, τ > 0.03), passive radiometers (τ > 0.1)." – It would be interesting to add figures (in Appendix?) of the BeCOOL mask color-coded with those OD thresholds corresponding to Figs. 1, 2b, 3b, and 4b. This would allow to better see the BeCOOL's enhanced detection capability. These cloud masks for the whole flights are displayed after our answers to RC3, the three case studies are highlighted with corresponding red numbers.

Line 249: "Optical and geometrical properties of mid- and high-level clouds"

Lines 250–251: "for all BeCOOL Level 2 profiles from the 2021 Strateole-2 campaign" – Does it represent more data than the 3 flights shown in previous sections? No, it does not. This sentence has been modified to remove any ambiguity. (Line **257**)

Lines 252–253: "All clouds with a reported base altitude below 5 km have been removed from both datasets to focus on free-tropospheric and TTL clouds." – Why this is not directly a condition for the cloud classification (Sect. 2.1)? The scene classification presented here has been done in a descriptive purpose. As this is the first article reporting BeCOOL's observations, we wanted it to be somehow exhaustive and not limited to the TTL, even if the following analysis focus on the TTL.

Lines 254–255: "Unlike the previous section, we use here the actual lidar-determined top and base altitude instead of the modified altitudes used for the cloud classification." – I don't understand the rationale for this. See comment for Lines 81–83. See previous answer about the cloud top/base altitudes.

Fig. 5b: It shows that 82 % of the lidar profiles show clouds with OD > $2·10^{-3}$. In Table 4, it was 75 %. What explains that the percentage is now higher? I might have understood if the value was lower since you removed mid-level clouds with base below 5 km from datasets (for some reasons I don't understand). Could it be due to the fact you don't use the same definition for cloud top and base altitudes in both sections (for some reasons I don't understand)? Those figures have changed a bit since we reprocessed the data but the point here is that Fig.5b reefers to profiles with clouds above 5 km while Table 4 only reefers to profiles with cirrus cloud, i.e. above 10 km.

Line 269: "and between 17 and 17.5 km for the base."

Lines 279–282: "For CALIOP, according to Yorks et al. (2011), multiple scattering effects due to the large size of the lidar footprint tend to lower the apparent cloud base altitude, enhancing the apparent geometrical depth of the cloud. This explains the differences between base altitude distributions and geometrical depth, while top altitude distributions show an excellent agreement." – Actually, this effect is mainly due to a non-ideal transient response of the PMTs (Lu et al. 2013, 2020). The effect is more visible in liquid clouds. I suspect the "Geometrically thin (a few hundred meters), horizontally extensive mid-level clouds are often found above, below 10 km, mainly between 5 and 8 km; they typically have large backscatter and are likely pure liquid or mixed-phase clouds." (Lines 68-70), to play a significant role in the geometrical depth difference you observed. Another reason for this effect comes from the difference in horizontal averaging in the detection algorithms. Compare to the detections performed at ~5-km horizontal resolution in the BeCOOL data, the detections performed at 20- and 80-km horizontal resolution in the CALIOP data can introduce a low bias in the cirrus cloud base retrieval when the cloud base altitude fluctuates at smaller scales. This does not affect the cirrus cloud tops very much are they appear flatter at those scales.

- Lu, X., Hu, Y., Liu, Z., Zeng, S., and Trepte, C.: CALIOP receiver transient response study, SPIE Optical Engineering + Applications, San Diego, California, United States, https://doi.org/10.1117/12.2033589, 2013.
- Lu, X., Hu, Y., Vaughan, M., Rodier, S., Trepte, C., Lucker, P., and Omar, A.: New attenuated backscatter profile by removing the CALIOP receiver's transient response, J. Quant. Spectrosc. Radiat. Transfer, 255, 107244, https://doi.org/10.1016/j.jqsrt.2020.107244, 2020.

Thank you very much for this enlightenment. The references have been added and the text changed accordingly. (Lines **286-290**)

Line 289: "GPS-RO" – Define here (where first mentioned).

Table 5: Mention in the caption this is for BeCOOL observations.

Line 336: "couldcloud base"

Lines 338–339: "Ultrathin TTL cirrus, with optical depth below the detection threshold of CALIOP ($\tau < 2 \cdot 10^{-3}$), are reported in 16 % of the lidar profiles" – Is 16 % coming from Table 4? My understanding is that this value corresponds to profiles containing ultrathin TTL cirrus only. In Sect. 4.2, you mentioned that clouds with $\tau < 2 \cdot 10^{-3}$ appear in 26 % of the profiles, but this is not limited to TTL cirrus. I would guess that the ultrathin TTL cirrus are detected in the lidar profiles with a fraction between 16 % and 26 %.

Indeed, you are right. The Ultrathin TTL cirrus are actually reported in 23 % of the profiles. A dedicated column had been added to table 5 to display these TTL cirrus cover percentages.

Fig. A2: "for case study 2"

**RC2**
Review of Observations of Tropical Tropopause Layer clouds from a balloon-borne lidar by Lesigne et al.

General: This study reports on measurements of cirrus clouds using the new balloon-borne microlidar BeCOOL, operated during three flights onboard superpressure balloons as part of the Strateole-2 measurement campaign. From three collocated measurements of cirrus with different microphysical properties, BeCOOL was compared to CALIOP. The agreement between the instruments was very good and, moreover, BeCOOL was found to be significantly more sensitive to thin cirrus compared to CALIOP. A significant finding of the study is that a comparison of the frequencies of occurrence of cirrus with various optical depths reveals that CALIOP misses ~20% of the cirrus, all within the range of $\tau < 2 \cdot 10^{-3}$. Furthermore, all BeCOOL cirrus observations are statistically analyzed for different cirrus types in different regions and, in addition, TTL cirrus top heights and thicknesses, classified according to optical depth.

This is an excellent and exciting study, presenting new insights in the properties and distribution of high altitude tropical cirrus clouds, based on high quality observations from a new instrument. The manuscript is well organized, fluently written and the figures are appropriate. It was a pleasure to read and review this article.
I have only a few minor comments, which are listed in the specific comments, that I would recommend to consider before publishing.

There is, however, one point on which the authors might have a second thought. To my feeling, the study sells itself a bit short - this is outlined in more detail in the specific comments to the abstract and title.

Specific comments:

Abstract: - I would include a sentence on the goal of the study after the first sentence, see ACP guidelines for authors:
https://www.atmospheric-chemistry-and-physics.net/policies/guidelines_for_authors.html
A sentence has been included to introduce the challenge of estimating TTL cirrus coverage regarding their wide range of optical depth. (Lines **3-5**)

- to my opinion it would be important to mention here that ~20% of the cirrus with $\tau < 2 \cdot 10^{-3}$ (cloud depth < 400m, cloud altitude > ~16km), which are mostly TTL cirrus, are not detected by CALIOP.
Would it be going too far to conclude that cirrus radiative-climate feedback estimates may therefore need to be reconsidered?
This is a good question. Figure 6 from Wang & Fu (2021) shows the gap in Longwave cloud radiative heating in the tropics between observations from the A-Train and reanalysis (ERA5 and MERRA2). They attribute this discrepancy to TTL thin cirrus. Such thin clouds may lack in reanalyses. Is the radiative effect of the ultrathin TTL cirrus reported here enough to reconcile observations and reanalyses ? We can not really conclude for now but this radiative impact is currently investigated, as stated among the perspectives at the end of the paper.

Wang, M., & Fu, Q. (2021). Stratosphere-troposphere exchange of air masses and ozone concentrations based on reanalyses and observations. Journal of Geophysical Research: Atmospheres, 126, e2021JD035159. https://doi.org/10.1029/2021JD035159

You have room to extend the abstract, it currently has 176 words and can be up to 250 words-

Title : Based on your exciting findings (and looking into the ACP guidelines for the title), you might think about changing the title, e.g. to something like:
'Observations of an unexpectedly/surprisingly high portion of Tropical Tropopause Layer clouds from a balloon-borne lidar'
Thank you for this suggestion, the title is now: ”Extensive coverage of ultrathin Tropical Tropopause Layer cirrus clouds revealed by a balloon-borne lidar”.

Line 32ff: For space-borne lidar observations, it might be worth to cite Sourdeval et al. (2018).
This reference has been added (line **36**)

Line 36ff: For airborne measurements of cirrus including TTL observations, Krämer et al. (2020) could be added.
This reference has been added (line **40**)

Line 56: 'fiels campaign' → field campaign

Line 71: 'The clouds' vertical structure can be fully resolved up to an optical depth τ max ≃ 3, ... ' Later, in Figure 10, optical depth up to 10 are shown ?
You are absolutely right, it was a mistake. Most of those potential outliers have been removed by the reprocessing, we still reports a few clouds with an optical depth about 4.

Line 173 'for such case' → for such a case

Line 315 '... placing almost half of the clouds in the wave phase in which both T ' and dT ' /dz are negative.' If I understand it right, these are the conditions of cirrus formation and the other cases represent aged cirrus, where only the longer living cirrus are found, or ? If this is true, it could be mentioned in the discussion of this result in the following paragraph.
This analysis have been done in a diagnostic purpose only, to confirm results from previous studies. As the collocation criterion is quite loose, and the GPS-RO temperature profiles smoothed on the horizontal, we are sure not to fully resolve the temperature anomaly field. We should stay cautious with the interpretation of the results in terms of processes without further analysis. If the wave phase in which both T' and dT'/dz are negative is indeed favorable for cirrus formation, Podglajen et al. (2018) suggest that ice crystals lifetime is also longer in this phase.

Table 3, caption: 'BeCOOL main profile classification, percentages of 10 minutes averaged profiles. Details on this classification can be found in Sect. 2.1.'

References:
Sourdeval, O., Gryspeerdt, E., Krämer, M., Goren, T., Delanoë, J., Afchine, A., Hemmer, F., and Quaas, J.: Ice crystal number concentration estimates from lidar–radar satellite remote sensing – Part 1: Method and evaluation, Atmos. Chem. Phys., 18, 14327–14350, https://doi.org/10.5194/acp-18-14327-2018, 2018.

Krämer, M., Rolf, C., Spelten, N., Afchine, A., Fahey, D., Jensen, E., Khaykin, S., Kuhn, T., Lawson, P., Lykov, A., Pan, L. L., Riese, M., Rollins, A., Stroh, F., Thornberry, T., Wolf, V., Woods, S., Spichtinger, P., Quaas, J., and Sourdeval, O.: A microphysics guide to cirrus –

Part 2: Climatologies of clouds and humidity from observations, Atmos. Chem. Phys., 20, 12569–12608, https://doi.org/10.5194/acp-20-12569-2020, 2020.

**RC3**

In this paper, the authors present observations made from microlidars operating from stratospheric balloons during three launches that were part of the Strateole-2 observation campaign. They compare these observations with colocated and simultaneous observations from the spaceborne lidar CALIOP. As the balloons move relatively slowly, lidar profiles can be accumulated for relatively long periods over the same atmospheric target, which with the proximity to the clouds significantly enhances the SNR of observations compared to a satellite lidar. The results show this enhanced sensitivity enables the detection of optically very thin cirrus clouds (optical depths < 2 10-3), well beyond CALIOP's detection abilities. These cirrus were ubiquitous near the TTL in the microlidar observation dataset.

This is a very well-written and impactful paper. The results are new, interesting and important. I have a few questions and minor comments below.

**Minor Comments**

l. 44: "significantly higher signal to noise ratio in the TTL" When such statements are made comparing the SNRs from BeCOOL and CALIPSO, it would be nice to clarify each time that the improvements are due to the accumulation time and proximity to the clouds. In absence of such precisions, a hasty reader could assume that BeCOOL has significantly higher SNR than CALIOP *in similar operating conditions*, which is probably not the case. These precisions are provided in the conclusion (l. 327) but they should also be added to the introduction.
This has been taken care of. (lines **47-48**)

l. 56: daytime observations are not mentioned, which is surprising given the attention to the improved SNR. Could you clarify the reasons behind this?
No measurements are made during daytime since the thermal conditions onboard the gondola do not allow the lidar to operate in good conditions. The laser diode experiences a spectral drift at high temperature that pushes its wavelength outside of the receiving filter window. It is visible on fig. 2 that the noise level is larger at the beginning of the night when temperature is still high in the gondola and decreases as the conditions in the gondola approach the optimal operating range.

l. 56: "fiels" > "field"

l. 57: What happens to the microlidars at the end of the flights? Are they lost?
Indeed, at the end of the flights the microlidars are lost. The gondolas fall below a parachute, usually ending in the ocean.

l. 82:"15% of the cloud optical depth lay above the new top altitude": Here I have failed to understand what has been done, and why. Could you clarify?

We finally kept a unique definition of cloud top and bas altitudes. See second answer to RC1 for details.

table 2: "wavelenght"

table 2: "CALIOP's 1064nm channel has a low SNR…" I was not aware of that, could you provide a reference? (Also line 193)
Here is an article assessing the performance of CALIOP after the three first years of operation. At that time, they report a 35% loss of SNR in the 1064 channel over the three first years due to a the receptor aging from exposition to radiations. The 532 channel is not subject to this effect as the receptor technologies are different (avalanche photodiode for 1064 channel, photomultiplier tubes for the 532 channel). The Strateole-2 campaign took place 14 years after CALIPSO's launch, the SNR of the 1064 channel is expected to have drop from a significant factor within this time.
Also we do not use the 1064 channel as we mainly rely on CALIOP L2 product where clouds are detected from the 532 channel. We removed any reference to the 1064 channel's low SNR since quantifying this is out of the scope of our study.

Hunt, W. H., D. M. Winker, M. A. Vaughan, K. A. Powell, P. L. Lucker, and C. Weimer, 2009: CALIPSO Lidar Description and Performance Assessment. *J. Atmos. Oceanic Technol.*, **26**, 1214–1228, https://doi.org/10.1175/2009JTECHA1223.1.

l. 128: "as been degraded"

l. 127-129: why degrade CALIOP's resolution horizontally, but not vertically? Averaging from 30 to 60m below 8.2km would get rid of the vertical step visible on Fig. 2. Also, if the goal is to display lidar curtains from both instruments at a comparable resolution, why not regrid BeCOOL's profiles vertically on a 60m resolution? I guess I don't understand very well the re-averaging choices made here.
You are right, the re-averaging choices are not consistent with the exposed motivation: displaying the curtains "at a comparable resolution". As the purpose of these curtains is purely illustrative, and no quantitative analysis are made from the re-averaged data, we removed this sentence (lines **144-145**). CALIOP's vertical resolution has been set to 60m above 8.2 km to hide most of the vertical step, which is an instrumental artifact and not the part we want the reader to focus on. Moreover we added in appendix a figure displaying lidar profiles from both instruments around the first coincidence, the differences in resolution and noise level appear clearly on this new figure.

Fig. 2 and others: the vertical black bars make the BeCOOL results hard to decipher. They make the CALIOP results appear of higher signal quality, which should definitely not be the case. Low-signal areas are particularly difficult to evaluate, which is a pity. Could the black bars be made thinner here, as in Fig. 1? Or maybe set to white/grey?
BeCOOL nighttime operating cycle is 10 minutes ON / 10 min OFF. The vertical bars, denoting these 10 min OFF legs, can not be made thinner without distorting the time scale. Following your suggestion, we set them to grey to differentiate "low signal" and "no signal".

Sect. 3.1: As in this case the BeCOOL/CALIPSO colocation is perfect, I would be very curious to see a superposition of the profiles measured by both instruments at the same time and location. The scene being sampled (a high optically thin cirrus + more opaque clouds below + clear sky) is quite rich and would give a good idea of the signal performance that can be reached from both instruments in the same conditions. This is such a unique situation and exceptional measurements that in my opinion it is worth spending a little more time on it.

You are right, we developed this case study adding in appendix a figure with the profiles, which is also convenient to discuss differences in noise level and performance.

l. 147: "thicken" > "thickens"

l. 155: "above above"

l. 155: why don't you show the total column optical depth, instead of showing only the optical depth above 10km?

We only show the optical depth above 10 km for two reasons :
- BeCOOL is designed to target cirrus cloud and the retrievals are more reliable on the upper part of the troposphere, as the SNR decrease towards the surface.
- On this particular scene, CALIOP's cloud detection fails and reports a cloud extending from the surface to 7 km, which is however flagged as " low/no confidence". On the following figure showing CALIOP's Vertical Feature Mask the coincidence is highlighted with a thin vertical red bar. The wrong cloud detection appears as a vertical magenta bar between the surface and 7 km. Comparing the total column optical depth would not be fair in this condition.

[Figure]

(downloaded from CALIPSO's website :
https://www-calipso.larc.nasa.gov/products/lidar/browse_images/show_v451_detail.php?

s=production&v=V4-51&browse_date=2021-11-29&orbit_time=14-34-28&page=3&granule_name=CAL_LID_L1-Standard-V4-51.2021-11-29T14-34-28ZN.hdf)

l. 158: "both retrieval"

Sect. 3.2: In the second case study, optical depths from both instruments agree very well (4 10-3 vs 3.8 10-3), in presence of an extremely optically thin cloud. How do you explain that the agreement appears much worse in the first case study (0.6 vs 0.4), which benefits from a much better colocation/coincidence and a more opaque, easier to detect cloud? Do you think you would get a much better agreement if you reprocessed the CALIOP data in the second case study, as you did with the first?

For the first case study, the agreement between retrieved optical depths from both instruments have been improved by BeCOOL's reprocessing (BeCOOL optical depth was 0.6 and is now 0.32, the closest CALIOP's profile reports 0.37)
The coincidence in time/space is almost perfect, unfortunately it happened between to cycles of BeCOOL observations. The median time of the closest 10 min profile from BeCOOL is 7 minutes before the satellite overpass, the next one is 15 minutes after. Those two profiles are shown on the left panel of the following figure, the three consecutive CALIOP 5 km profiles closest to coincidence are shown on the right panel. Shaded areas around the profiles represent the min/max envelope of the averaged profiles (10 x 1 min profiles for BeCOOL, 15 x 0.333 km profiles for CALIOP). The high spatial variability within this clouds clearly appears on this figure : CALIOP's profiles are quite different one from another. This high variability, also appearing on both lidar curtains of fig. 02 of the article, explains the discrepancies in retrieved optical depths.
On the contrary, for the second case study, the thin cirrus cloud is spatially very homogeneous, allowing to average CALIOP's profiles over large horizontal legs (80 km) to retrieved optical depths where the cloud was not reported in CALIOP L2 files.
Since BeCOOL's reprocessing has improved the accuracy of retrieved optical depth, the values compare quite well: 0.37 for CALIOP, 0.32 for BeCOOL.

**First case study : lidar profiles**

[Figure]

l. 186: "We can attempt to estimate the horizontal extension of this UTTC assuming an horizontal extension…" could you please rephrase this? It reads as if you estimate the extension by assuming an extension. Also, what supports the assumption that the cirrus expands a few hundred kms in any direction?

This sentence has been rephrased.
In this case study, as it can be seen on the lidar curtains, the cirrus clearly extends a few hundreds of kilometers along both instrument tracks, although those tracks are not orthogonal. Winker & Trepte (1998) documented laminar cirrus around the tropopause from LITE lidar, reporting a mean latitudinal extension of 500 km, with a maximum of 2700 km. CALIOP's observation of this thin cirrus are in line with these results. On the other hand, despite the strong variability in flight direction, the balloon's track are on average zonally oriented, giving a longitudinal point of view. During the APE-THESEO campaign (Peter et al., 2003) an ultrathin cirrus was observed from aircraft and sample in several directions, showing an horizontal extension of several thousands of square kilometers. Our rough estimation was done to be compared with this results, the orders of magnitude are the same even if, indeed, the assumptions are quite strong.

Table 4 and the following paragraph: These results make me wonder if the cirrus cover is only limited by the instrument's sensitivity, i.e. if there is an optical depth continuum. Are

there reasons to believe that an instrument with an even finer sensitivity to optically thin clouds wouldn't report even higher number of cirrus cover than BeCOOL, and detect super-super-thin cirrus everywhere? If you plot the cirrus cloud cover as a function of the instrument's optical depth sensitivity, would the cloud cover be 100% at the origin?

Here is the figure showing the cirrus coverage as a function of optical depth sensitivity inferred from both BeCOOL and CALIOP's observations. The logarithmic scale on the x-axis makes any extrapolation down to the origin quite hazardous and no clear conclusion can be drawn from this single figure. One could however speculate that the coverage would reach a plateau lower than 100%.

[Figure]

We are confident that an instrument with a finer sensitivity would report a larger cirrus coverage, in that sense BeCOOL's observations support the existence of the clear sky-cloud optical depth continuum highlighted by Balmes and Fu (2018). But the cirrus coverage also depends on the amount of water vapor available for cloud formation and we would still expect some regional contrast between convective regions such as the Maritime Continent, and dryer regions away from convection.

I know this is quite tricky, but could you discuss a little bit if you think the optically very thin cirrus that are so frequent in the microlidar dataset are a generic/persistent feature of the regions you sample? Of other regions/periods? Previous studies such as Balmes and Fu 2018 suggest that similar optically-very-thin cirrus are found even in the extratropics and

over land. Could you contrast your results with those? Doing so could help generalize your results to longer time periods and/or larger spatial scales.

The optically very thin TTL cirrus are quite comparable to the laminar cirrus reported by Winker & Trepte (1998) from 10 days of LITE observations in September 1994. Those clouds are only detected within the tropics (in a broad sense: from 35°N to 20°S), near the tropopause, over land as well as over ocean, but not omnipresent. Balmes & Fu (2018) documented very thin ice clouds (optical depth < 0.01) from two ground-based Raman lidars located respectively at latitudes of 36.6°N and 12.3°S, which is consistent with Winker & Trepte (1998) observations. BeCOOL's observation are closer to the equator, limited to a 5°N – 10°S latitude band.

Wang et al. (2019) studied laminar TTL cirrus from ten years of reprocessed CALIOP's L1 data and show that those clouds follow the same climatology (main geographical distribution patterns on their fig.5, seasonal variations fig.6) than the total cirrus (including anvil cirrus). Their reprocessed laminar cirrus optical depth distribution (their fig.3) extends lower than the $2.10^{-3}$ cut-off appearing in CALIOP's L2 data, however it does not extend as low as BeCOOL's. We have no observational evidence that ultrathin cirrus reported from BeCOOL do not to follow the same climatology, established from CALIOP for thicker cirrus. Eventually, to generalize those results, we hope that during the next Strateole-2 campaign (scheduled for the 2025-2026 boreal winter) the balloons carrying BeCOOL will achieve circumnavigation to be able to document the ultrathin cirrus cover all around the equator belt, in particular over continents. Moreover, BeCOOL microlidar should be deployed during future campaigns outside of the tropics, in particular for a transatlantic flight at high latitudes and over the arctic region. These perspectives have been added at the end of the article. (lines **349-353**)

Next two pages :

BeCOOL cloud mask color-coded with the optical depth thresholds, red numbers highlight the three case studies.

**Flight 1**
**2021-10-20 to 2021-10-31**

[Figure]

**Flight 2 (panel 1/3)**
**2021-11-05 to 2021-11-22**

[Figure]

**Flight 2 (panel 2/3)**
**2021-11-23 to 2021-12-10**

[Figure]

[Figure]

**Flight 2 (panel 3/3)**
**2021-12-11 to 2021-12-28**

[Figure]

[Figure]

surface    noise    opaque    clear

0        0.002      0.03      0.1

optical depth

[Figure]

**Flight 3 (panel 1/4)**
**2021-11-16 to 2021-12-02**

[Figure]

**Flight 3 (panel 2/4)**
**2021-12-03 to 2021-12-20**

[Figure]

**Flight 3 (panel 3/4)**
**2021-12-21 to 2022-01-07**

[Figure]

**Flight 3 (panel 4/4)**
**2022-01-08 to 2022-01-25**